# Sim2Real VLA: Zero-Shot Generalization of Synthesized Skills to Realistic Manipulation

**Runyi Zhao[1], Sheng Xu[1], Ruixing Jin[1], Yueci Deng[1], Yunxin Tai[2], Kui Jia[1,2], Guiliang Liu[1,3]***

[1]School of Data Science, The Chinese University of Hong Kong, Shenzhen
[2]DexForce Technology, [3]Shenzhen Loop Area Institute.
`{runyizhao,shengxu1,ruixingjin,yuecideng}@link.cuhk.edu.cn`
`yunxintai@dexforce.com`
`{kuijia,liuguiliang}@cuhk.edu.cn`

## Abstract

Vision-Language-Action (VLA) models represent a critical milestone toward embodied intelligence in robotic manipulation. To support their training, recent research has developed high-performance simulation engines for data synthesis. However, their effectiveness is still significantly limited by the simulation-to-reality (Sim2Real) gap, as policies trained on synthetic data often fail to generalize reliably to the real world. To address this challenge, we present Sim2Real-VLA, a generalist robot control model trained exclusively on synthetic data, yet capable of transferring seamlessly to real-world manipulation tasks. Sim2Real-VLA features a dual-system architecture: a high-level planner that infers chains-of-affordances, and a low-level actor that executes and validates these plans in real time via a tokenized action space. This design filters out manipulation-irrelevant features and prioritizes motion-critical dynamics, thereby enhancing Sim2Real domain transfer. Besides, a notable advantage of Sim2Real-VLA lies in its tight integration with automated data generation for manipulation skills, eliminating the need for manual fine-tuning and enabling scalable, hands-free training. Empirical evaluations across bimanual, dexterous, and long-horizon tasks show that Sim2Real-VLA consistently outperforms previous VLA baselines under diverse real-world environments and domain shifts. The source code is available at `https://github.com/DexForce/EmbodiChain`.

## 1 Introduction

Designing precise and scalable robotic manipulation policy represents a key milestone toward realizing artificial general intelligence (AGI) (Smith et al., 2012). Despite significant advances in hardware design, control algorithms, and simulation platforms, traditional robotic systems remain highly specialized, often requiring task-specific engineering and extensive manual tuning. Recent advances in large foundation models provide a promising pathway to address these limitations with the development of generalizable manipulation policies (Zhao et al., 2023b; Zhang et al., 2024). The Vision-Language-Action (VLA) models, which integrate visual observations, natural language commands, and robotic control actions, have emerged as a prevailing architecture for implementing generalist agents in real-world robotic applications (Ma et al., 2024b; Zheng et al., 2025).

As a foundation model for robotic control, VLA training typically follows a data-driven pipeline. This process demands a large amount of robot-operating data, the collection of which involves intensive manual effort and access to specialized hardware. While previous studies have demonstrated the effectiveness of VLA pre-training on internet videos (Luo et al., 2025) and cross-embodiment datasets (O'Neill et al., 2024), deploying these models typically requires additional rounds of fine-tuning on the target robot and task-specific skills. To enable more scalable training, recent studies have explored the use of synthesized or simulated data (Mandlekar et al., 2023; Deng et al., 2025; Liu et al., 2025a). A key advantage of such data is that it can be generated at a large scale using high-performance computing clusters via automatic skill acquisition (Nasiriany et al., 2024; Wang et al., 2024b; Mu et al., 2024; Hua et al., 2024; Zhao et al., 2024). Nevertheless, models trained exclusively on these datasets are subject to a Sim2Real domain gap when deployed in practice.

---

*Corresponding author: Guiliang Liu, liuguiliang@cuhk.edu.cn.

To close this domain gap, mainstream research has focused on developing photo-realistic and physics-accurate simulation environments (Hua et al., 2024; Puig et al., 2024) or world models (Agarwal et al., 2025). However, accurately modeling real-world dynamics remains a significant challenge that has yet to be solved (Bharadhwaj, 2024). More importantly, recent studies (Xie et al., 2024; Liu et al., 2025a) have shown that factors such as lighting conditions and background textures, despite consuming substantial modeling resources, are essentially irrelevant to manipulation performance. These findings call for an alternative approach: instead of focusing on generating high-fidelity data, we propose addressing the Sim2Real by redesigning the VLA architecture.

In this study, we introduce Sim2Real-VLA, which, despite being trained solely on synthetic data, demonstrates generalizable and sustained manipulation performance across diverse real-world environments. To address the enduring domain gap between synthesized and realistic data, Sim2Real-VLA integrates a generalization mechanism in model design. Specifically, Sim2Real-VLA utilize a dual system design, encompassing a high-level planner and lower-level actor, interconnected by affordance signals in finishing the given task. Such affordances play a fundamental role in robotic manipulation within our Sim2Real-VLA design because: 1) The key steps in executing a long-horizon task can be abstracted as a chain of affordances, thereby providing a structured basis for embodied reasoning within *the planning system*; 2) The predicted affordance signals function as critical supervision for producing robot control outputs within *the acting system*; and 3) Contemporary methods for *manipulation skill acquisition* (Ma et al., 2024a; Mu et al., 2024; Wang et al., 2024b) are grounded in affordances, which can be consistently derived in simulation environments[1] and provide supervision signals for training Sim2Real-VLA. By coupling reasoning and acting with manipulation affordances derived from object-oriented observations, Sim2Real-VLA filters out manipulation-irrelevant features and concentrates on task-relevant dynamics, thereby effectively closing the Sim2Real gap.

We conduct extensive studies across multiple tasks involving bimanual, dexterous, and long-horizon manipulation. Our findings reveal that even state-of-the-art VLA models, when trained solely on synthesized data, struggle to perform effectively in real-world manipulation scenarios. In contrast, Sim2Real-VLA outperforms baselines with a minimal Sim2Real gap, achieving over 35% higher success in realistic environments. More importantly, our quantitative experiments demonstrate that its zero-shot Sim2Real capability generalizes reliably across a wide range of domain shifts.

## 2 RELATED WORKS

**VLA Model for Robot Manipulation.** In recent years, VLA models have emerged as a prevailing paradigm in multi-modal foundation models and have been successfully applied to robot control tasks (Ma et al., 2024b; Firoozi et al., 2024). Among these tasks, a critical application is dexterous manipulation, which requires the model to comprehend the given commands, interact with various objects, and dynamically respond to changing environments (Zheng et al., 2025). Building upon the VLM backbone (Zhang et al., 2024), earlier VLA models, such as OpenVLA (Kim et al., 2024), Otco (Ghosh et al., 2024), RTs (Brohan et al., 2023; Zitkovich et al., 2023), RDT (Liu et al., 2025b), and $\pi_0$ (Black et al., 2024), typically utilized an end-to-end model architecture. To enhance the efficiency of policy inference, recent approaches, such as HelixFigure AI (2025), Gr00NtBjorck et al. (2025), Gemini (Team et al., 2025), AgiBot (Bu et al., 2025), and other dual-system frameworks (Shentu et al., 2024; Han et al., 2024; Bu et al., 2024; Chen et al., 2025; Wen et al., 2025; Zhang et al., 2025), have adopted a two-level architecture. This design comprises a low-frequency VLM system responsible for semantic understanding and embodied reasoning, alongside a high-frequency policy model that efficiently predicts control signals at a faster rate. Despite these advancements, training such models often demands collecting real-world datasets and fine-tuning the VLA models for specific robotic tasks (Nasiriany et al., 2024; Mu et al., 2024).

**Sim2Real Generalization.** In the application of robot Manipulation, Sim2Real generalization techniques often involve: 1) *Domain randomization* tackles Out-of-Distribution (OoD) scenarios in practical applications augmenting the training dataset with randomized visual and physical features (Chen et al., 2022). To determine the scale of randomization, recent studies have determined relevant parameters by automatic learning (OpenAI et al., 2019), active exploration (Mehta et al., 2019), Bayesian update (Muratore et al., 2021b;a), offline inference (Tiboni et al., 2023) and continual

---

[1]The simulation system provides complete access to the spatial, physical, semantic, and morphological properties of objects, which substantially facilitates accurate affordance estimation.

learning (Josifovski et al., 2024). 2) *Domain adaptation* reduces the gap between simulated and real-world domains by aligning them within a shared feature space, such as 2D images (Bousmalis et al., 2018; Zhang et al., 2019), 3D point clouds (Lobos-Tsunekawa & Harada, 2020; Qin et al., 2022; Chen et al., 2023), or environmental dynamics (Memmel et al., 2024; Josifovski et al., 2025). 3) *Real2Sim* projection maps real-world scenes into the simulation environment, enabling the system to better capture realistic semantics (Villasevil et al., 2024; Liu et al., 2025a; Tai et al., 2025). However, it lacks principled studies on redesigning VLA models to close the Sim2Real gap.

## 3 PROBLEM FORMULATION

**Environment for Robotic Manipulation.** To formulate the problem of Sim2Real robot manipulation, we can formulate the real-world robot deploying environment into a Partially Observable Markov Decision Process (POMDP) $\mathcal{M} = (\mathcal{S}, \mathcal{A}, \mathcal{O}, P_\mathcal{T}, R, \mu_0, \gamma)$ where: 1) The state $s_t \in \mathcal{S}$ captures the semantic information of a scene, encompassing the configuration (e.g., layouts, appearance, and physical characteristics) of various types of objects and the robots. 2) The action $a_t \in \mathcal{A}$ indicates control signals for the target robot. We follow diffusion policies (Chi et al., 2023; Liu et al., 2025b), and utilize the joint angles in the robot for presenting actions. 3) The observation $o \in \mathcal{O}$ represents the perceptual signals captured by sensors. These observations are typically non-Markovian and provide only partial information about the current state. 4) Transition function $P_\mathcal{T}$ characterizes the impact of robot action $a_t$ to the configuration of current state $s_t$, thereby projecting the $s_t$ to a new scene represented by $s_{t+1}$. 5) The reward function $R(s, a)$ represents how effectively a robot completes a targeted task after taking action $a$ in state $s$ (Appendix A.2 introduces more details). 6) $\rho_0$ denotes the initial state distribution and $\gamma \in (0, 1]$ denotes the discounting factor which weights the importance of future rewards relative to immediate rewards.

**Zero-Shot Sim2Real Learning for Control Policy.** Within the robot's operational environment, our objective is to learn a control policy $\pi(a_t, \ldots, a_{t+M} \mid o_{t-H}, \ldots, o_t, l)$ that predicts a sequence of $M$ future actions $a_t, \ldots, a_{t+M}$ based on a history of $H$ past observations $o_{t-H}, \ldots, o_t$ and tasks annotation $l$ (Zhao et al., 2023a). At a time step $t$, $o_t$ captures both proprioception $o_t^p$ and visual signals $o_t^v$ from multi-view cameras. By leveraging observations and language-based task annotations, the control policy can be effectively instantiated as a VLA model (Kim et al., 2024). In this study, we consider learning a control policy under a Sim2Real transferable setting:

**Definition 3.1 (Sim2Real Transferable Policy Learning)** *Let $\mathcal{M}$ denote the real-world environment, and $\widehat{\mathcal{M}} = (\hat{\mathcal{S}}, \hat{\mathcal{A}}, \hat{\mathcal{O}}, \hat{P}_\mathcal{T}, \hat{r}, \hat{\mu}_0, \hat{\gamma})$ denote a corresponding simulated environment. Let $\pi$ denote the policy trained on a collection of skill trajectories $\hat{\tau} = [\hat{o}_0, \hat{a}_0, \hat{o}_1, \hat{a}_1, \ldots, \hat{o}_T, \hat{a}_T, l]$ generated in $\widehat{\mathcal{M}}$, where $\hat{o}_t \in \hat{\mathcal{O}}$, $\hat{a}_t \in \hat{\mathcal{A}}$, and $l$ is a task description such that $\pi = \arg\max_\pi \mathbb{E}_{\hat{\tau}}[\mathcal{J}(\hat{\tau}, \pi; l)]^2$. Nevertheless, we expect goal $\pi$ to achieve the optimal performance in the realistic environment $\mathcal{M}$.*

Notably, this Sim2Real transfer is conducted *in a zero-shot manner*, where no real-world demonstrations are used during training, but the learned policy must generalize to and solve real-world manipulation tasks. However, as demonstrated by prior studies (Nasiriany et al., 2024; Wang et al., 2024a), the discrepancy between the simulated environment $\widehat{\mathcal{M}}$ and the real-world environment $\mathcal{M}$ poses significant challenges (as also evidenced by our experiment results in Section 5.1). Directly using simulated skills $\hat{\tau}$ to supervise the training of real-world policies $\pi$ often leads to suboptimal performance due to this domain gap. Although domain randomization and adaptation methods have demonstrated successful examples in general robotic control (Mehta et al., 2019; Muratore et al., 2021a; Tiboni et al., 2023; Josifovski et al., 2024), their applicability to manipulation tasks in the context of VLA models remains an open question. This work aims to design a structured framework that enables learning a zero-shot Sim2Real transferable object manipulation policy.

## 4 VLA MODEL FOR ZERO-SHOT SIM2REAL GENERALIZATION

To bridge the Sim2Real gap, rather than constructing a more sophisticated data engine to replicate real-world dynamics, our study considers a *model-side* solution. This approach acts as a filter for realistic dynamics, focusing exclusively on features critical for manipulation, guided by the simulation

---

[2]We use $\mathcal{J}(\cdot, \cdot; l)$ to denote the objective parameterized by language command $l$.

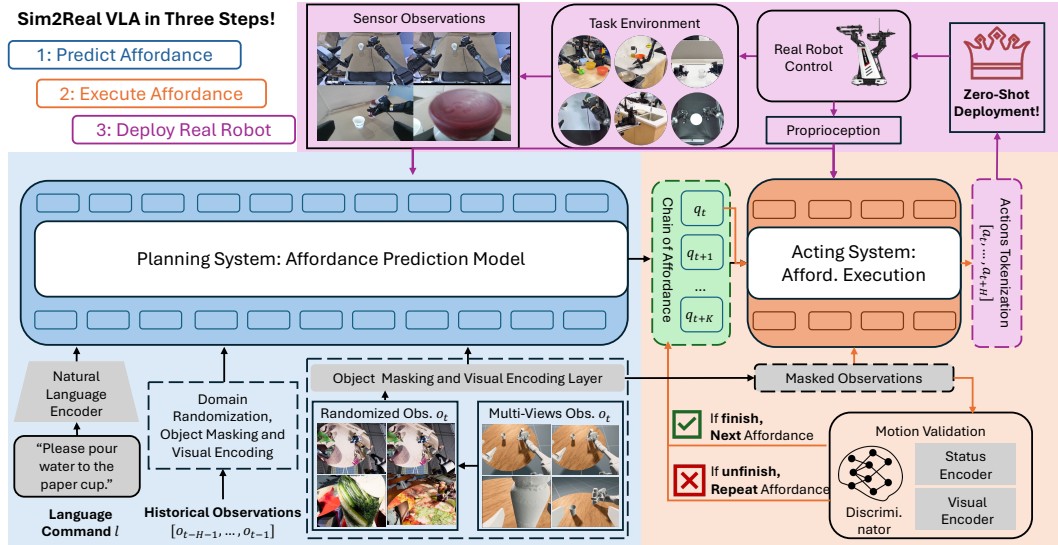

Figure 1: The pipeline of our Sim2Real-VLA model consists of two main components: a planning system ( Section 4.1) that enables embodied reasoning through a chain of affordances, and an acting system (Section 4.2) that drives the model by executing the planned affordances. This design allows the model to transfer seamlessly to real-world manipulation tasks.

engine. This strategy not only significantly reduces modeling complexity but, more importantly, enables the skills learned from simulated data to effectively drive the VLA model.

To implement the Sim2Real VLA model, we design an affordance-driven, object-oriented framework that integrates planning and acting, structured as follows:

## 4.1 PLANNING SYSTEM: ROBOTIC MANIPULATION AS CHAIN-OF-AFFORDANCE

The goal of the planning system is to reason about the essential steps for finishing the given tasks described by the language command $l$. Specifically, our planning system predicts a chain of affordance $[q_0, \ldots, q_K]$ to represent these essential robot motions. To support the characterization and prediction of affordance, we adapt the visual observation into the object-oriented representation and design an affordance reasoning process to finish the given tasks.

**Object-Oriented Adaptation in Observation Space.** Since some key waypoints in manipulation trajectories are inherently object-centric, synthesizing feasible manipulation trajectories requires access to each object's pose and morphology (e.g., position, orientation, and shape parameters). While such information is typically unavailable from real-world cameras, it can be recovered in simulation. Given a fixed camera configuration, we render observations as $\hat{o}_t = f_\xi(e_0, e_1, e_2, \ldots)$ where $e_i$ denotes an object (e.g., including those with articulated, rigid and soft body) in the simulated scene and $f_\xi$ is the rendering function parameterized by the semantic configuration $\xi$ (including layouts, camera view). Performing object-oriented adaptation is essentially learning the reverse of $f_\xi$ by minimizing the following negative log-likelihood loss function:

$$\mathcal{L}(\theta) = \mathbb{E}_{\hat{\tau}, [m_t^i]_{i,t=0}^{I,T}} \left[ -\sum_{t=0}^{T} \sum_{i=0}^{I} \log \left( p_\theta^R(m_t^i | o_t^\xi, \ldots, o_{t-H}^\xi) \cdot p^d(o_t^\xi | \hat{o}_t, \xi) \right) \right] \quad (1)$$

where $\hat{\tau}$ and $m_t^i$ denote the skill trajectory and object mask. As objects are not necessarily fully observable, we adopt a probabilistic predictor $p_\theta^R$ for object recovery, thereby accounting for the underlying uncertainty. Moreover, because training data are generated in simulation and rendered objects may deviate from real-world appearance, $p^d$ incorporates Domain Randomization (DR) into the observation $\hat{o}_t$ under the scene configuration $\xi$. Specifically, it performs the following operations

*1) Strategic DR Features Selection.* Sim2Real-VLA incorporates a large foundation model (e.g., a vision–language model such as GPT-5) into the domain randomization (DR) process. It leverages the model's reasoning ability to rank DR features and define their sampling ranges (see Table 1), using the task description, current observations $\hat{o}_t$, and the simulated environment configuration $\xi$. For

simplicity, we characterize $p^d$ as a joint uniform distribution over the selected features within their respective ranges (Mehta et al., 2019).

Table 1: The set of DR features for characterizing the Sim2Real generalization gap in robotic manipulation tasks (Xie et al., 2024).

| Level | Domain Randomization (DR) Features |
|---|---|
| Scene Level | Lighting, Table Texture, Background, Distractors, Object Locations, Object Orientations, Object Texture and Object Shape |
| Robotic Level | Cameras Position, Cameras Orientation, Cameras Field of View, Initial end-effector pose |

*2) Flows of DR in Sequence Observation.* At each time step, unlike previous methods that rely on a fixed set of randomized features to generate a trajectory Mu et al. (2024), $p^d$ develops a flow of DR by resampling action-invariant features, such as lighting, textures, and backgrounds, whose values have limited influence on the robot's actions. By performing DR at a higher frequency, our approach enhances the generalization capability of the learned policy. Figure 2 (below) shows an example of DR for the observations of water pouring motions.

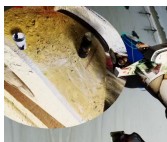 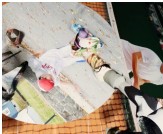 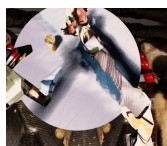 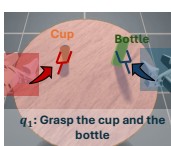 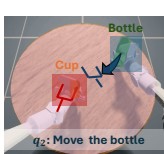 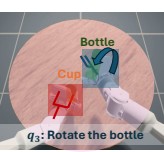

Figure 2: An example of the DR flow (left three images) and the chain of affordances (right three images) generated in the simulated environment for the water-pouring task.

**Reasoning via Chain-of-Affordance.** Conditioned on the visual observation and the instruction $l$, the planning system infers the appropriate chain-of-affordance $\boldsymbol{q} = [q_0, \ldots, q_K]$. Each $q_\tau$ corresponds to a series of geometrically structured keypoints that represent a key end-effector pose that governs the robot–object interaction necessary to accomplish an atomic task. By predicting $\boldsymbol{q}$, our planning system of Sim2Real VLA essentially reasons about the sequence of atomic tasks based on the provided instruction, alongside the robot's key motion in finishing these tasks. Unlike the embodied Chain-of-Thoughts (CoT) reasoning based on language description (Zawalski et al., 2024), Sim2Real VLA performs reasoning with affordances, thereby better aligning the planned atomic tasks with objects, robot configurations, and commands $l$. During implementation, the objective of Sim2Real VLA is to learn the predictive distribution $p^A_\phi(q_{k,t}, \ldots, q_{K,t} \mid \hat{m}_t, o^\xi_t, \ldots, o^\xi_{t-H}, l)$, where $\hat{m}_t$ represents the predicted mask of the target object in equation 1 and $q_k$ denotes the next affordance based on the current observation. To facilitate implementation, we decompose the joint distribution into conditional components and optimize the model by minimizing the following loss function:

$$\mathcal{L}(\phi) = \mathbb{E}_{\hat{\tau}, [q_k]_{k,t=0}^{K,T}} \left[ - \sum_{t=0}^{T} \left( \sum_{k=1}^{K} \log p^A_\phi(q_{k,t} \mid q_{k-1,t}, \hat{m}_t, o^\xi_t, \ldots, o^\xi_{t-H}, l) \cdot p^d(o^\xi_t | \hat{o}, \xi) \right) \right] \quad (2)$$

where $p^d$ is the aforementioned domain randomization function. Conditioning on the previous affordance, the target object, and visual observations, and the command, $p^A$ can perform in-time prediction of the future affordance to finish the given task, but at a frequency of muster smaller than that of the robot acting system 4.2. Figure 2 (right) illustrates the chain of affordances generated in the simulation environment.

## 4.2 ACTING SYSTEM: PREDICTIVE CONTROL AS AFFORDANCE EXECUTION

Upon receiving the predicted affordance sequence $[q_0, \ldots, q_K]$ from the high-level planning module, the low-level acting system $\pi_\omega(a_t, \ldots, a_{t+M} \mid q_k, \hat{m}_t, o_{t-H}, \ldots, o_t)^3$ iteratively executes each

---

[3] Here, $a_{t-1}$ denotes the final action taken to achieve the previous affordance $q_{k-1}$.

affordance by guiding the end-effector toward the designated target pose, and verifies at each step that the resulting motion successfully achieves the intended affordance.

**Affordance Execution.** To execute affordances both efficiently and accurately, the acting system leverages a tokenized action space and employs a decoupled estimation strategy for controlling the manipulation actions of a bimanual robot. This design enables the system to remain both flexible and modular while reducing unnecessary dependencies between the two arms.

*Arm-Decoupled Estimation.* In the decoupled estimation framework, the policy $\pi_\omega$ is split into two independent components, $\pi_{\omega_l}$ and $\pi_{\omega_r}$, which control the left and right arms of the robot, respectively. Although these models are jointly trained to complete bimanual manipulation tasks, they are independently parameterized. Each model only has access to its own relevant visual observations (from arm-mounted and top-down cameras) and to the specific affordance target associated with its arm. This independence is crucial for reliable affordance tracking, as it prevents the acting model from misattributing attention to the other arm's goals or state when computing its own actions.

*Tokenized Action Space.* Rather than operating directly in high-dimensional, continuous control space, we apply a frequency-domain representation named Discrete Cosine Transform (DCT) (Ahmed et al., 2006; Pertsch et al., 2025) to a normalized segment of continuous actions. This transformation converts the signal into the frequency domain, where we can efficiently compress it. The resulting DCT coefficients are then quantized, and we apply Byte-Pair Encoding (BPE) (Gage, 1994) to compress the sequence of per-dimension coefficients into a compact action token sequence, denoted as $a^{DCT}$. This tokenization significantly reduces the complexity of the action space, enabling faster and more robust learning while retaining the essential temporal and spatial characteristics of the original motion.

**Affordance Validation.** Since the chain of affordances is sequentially dependent, the acting model must successfully execute the earlier ones before implementing the later ones, especially in long-term tasks. To achieve this, we develop a validation model that dynamically determines whether the target pose has been tracked and whether the system should proceed to the next affordance. In this way, the acting system can be reactive to the failure case, which greatly improves the system's robustness.

### 4.3 Automatic Data Generation Pipeline for Sim2Real VLA Training

Since Sim2Real-VLA is trained exclusively on simulated data, its performance heavily depends on the efficiency and scalability of simulated data generation. Recent advances in agentic skill acquisition (Ma et al., 2024a; Wang et al., 2024b) have made it possible to generate such data automatically, without manual intervention or additional hardware, thereby enabling a more scalable, efficient, and cost-effective process for generating training data. This section introduces our automated data generation pipeline, highlighting the core components that can, in principle, generate the integration of Sim2Real-VLA with training environment construction, skill dataset generation, and the provision of relevant guiding signals.

1) **Real2Sim Data Projection** maps descriptive observations of the target task from real-world applications to the simulated environment. Following (Dai et al., 2024; Liu et al., 2025a), the project encompasses *static scene information*, including the orientation, position, and morphological features of objects as well as their spatial layout, and *dynamic action trajectories* derived from teleoperation and human demonstration videos. The resulting simulation environment faithfully preserves the semantic structure of both the tasks and their contexts. Check Appendix A.3 for more details.

2) **Generative Scene Scaling** samples diverse configurations within the simulated environment based on the Real2Sim prior information obtained from the aforementioned projection. The sampling process spans both scene-level and robot-level features in Table 1. Each sampled configuration defines a distinct scene, resulting in a set of candidate environments derived from the target scene. Figure 3 illustrates an example of the Real2Sim projection along with the corresponding environment scaling. The spatial and morphological information of objects is available in the simulation engines, enabling the generation of object masks $m_i$ during robot operation. These masks can serve as effective supervision signals in the objective (1). We present more details in Appendix A.4.

3) **Automatic Skill Acquisition** generates operation trajectories for accomplishing the target task. The embodied agent, equipped with a vision–language model, decomposes the task into atomic units (Nasiriany et al., 2024), identifies the target object from the input instruction (Fang et al., 2023),

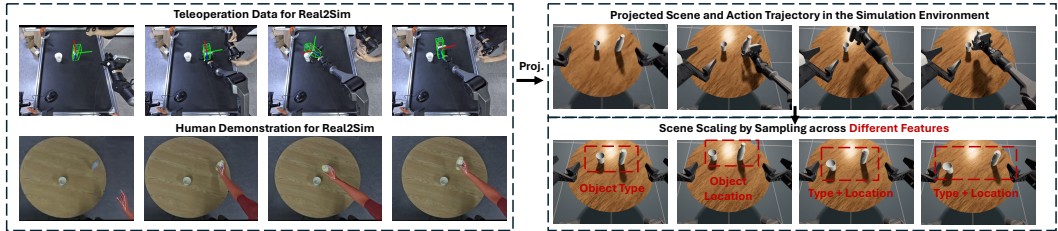

Figure 3: An example of our data generation pipeline, which projects scenes and action trajectories from heterogeneous sources (videos or teleoperation) into the simulated environment, and then scales the environment with diverse randomized features.

and produces candidate grasping and manipulation poses for the end-effector (Mu et al., 2024). From these key poses, a generalized inverse kinematics (IK) algorithm is applied to compute the corresponding joint angles required to drive the robot arms. The key poses at the end of each atomic task are utilized as an affordance supervision ($q = [q_0, \ldots, q_K]$ in the objective (2), which aligns the tracking of the affordance with finishing atomic tasks. The instructions of atomic tasks, the computed joint angles, the identified affordances, and the rendered images from the simulator are utilized as the training data for our Sim2Real-VLA model. Appendix A.5 shows more details.

## 5 EMPIRICAL EVALUATION

We conduct an extensive experimental study of Sim2Real-VLA, evaluating its performance from the following perspectives: 1) applicability to real-world manipulation tasks and 2) generalizability across diverse domain gaps. Additionally, we visualize the attention maps during the operation of robotic manipulation, demonstrating the rationality of Sim2Real-VLA during robotic manipulation. Our experiments are conducted based on the Agilex CobotMagic robot by following (Fu et al., 2024).

### 5.1 APPLICABILITY: TOWARD ROBUST SIM2REAL PERFORMANCE IN LONG-HORIZON TASKS

While prior studies have demonstrated zero-shot Sim2Real performance in short-term tasks such as object grasping and placing (Collins et al., 2019; Xie et al., 2024; Deng et al., 2025), the extent to which this performance generalizes to long-horizon manipulation remains largely unexplored. In this study, we evaluate our method using the manipulation tasks summarized in Table 2. For each task, we generate training data in the simulation environment (Section 4.3), learn a manipulation policy from the synthesized data, and evaluate its generalization performance across different environments.

Table 2: Task descriptions with decomposed action steps and arm type.

| Task | Steps | Arm Type |
|------|-------|----------|
| Single-Arm Water Pour | (1) Grasp bottle → (2) Move bottle to cup → (3) Pour water → (4) Return bottle | Single-Arm |
| Dual-Arm Water Pour | (1) Grasp bottle → (2) Grasp cup → (3) Move bottle to pouring position → (4) Move cup to receiving position → (5) Pour water → (6) Return cup → (7) Return bottle | Dual-Arm |
| Table Rearrangement | (1) Grasp fork → (2) Place fork beside plate → (3) Grasp spoon → (4) Place spoon beside plate | Dual-Arm |
| Items Hand-Over and Place | (1) Grasp an item → (2) Transfer the item to the other hand in the air → (3) Place the item into the holder | Dual-Arm |
| Basket Pick-and-Place | (1) Grasp an item → (2) Place the item into basket → (3) Grasp basket → (4) Place basket down | Dual-Arm |
| Pan Open-Pick-and-Place | (1) Grasp lid → (2) Open lid → (3) Grasp an item → (4) Place the item into pot → (5) Close lid | Dual-Arm |

**Evaluation Metrics.** We evaluate the robotic manipulation policy in both simulation (**Sim.**) and real-world (**Real.**) environments. To prevent overfitting and to examine generalization performance, we sample different feature configurations from Table 1 in both environments at each run. *The simulation environment (**Sim.**) incorporates a certain level of randomness* to better reflect the variability and

uncertainties the robot may encounter in real-world scenarios. In this way, simply replaying training data is insufficient to achieve strong performance in both simulation and real-world environments. We report the number of runs, the times of success manipulation, and the average number of steps required to complete the tasks in the real-world environment. Additionally, to reflect the efficiency of robotic manipulation, we also report the average number of **steps** required to complete each task, as more effective policies tend to complete tasks in fewer steps. For unsuccessful trials where the robot fails to complete the task, we report the predefined maximum step limit as an upper bound.

**Baselines.** To benchmark our Sim2Real-VLA , we compare it against several representative baselines, including 1) **Action Chunking with Transformers (ACT)** (Zhao et al., 2023a) leverages sequence modeling to learn temporally extended action policies, 2) **Diffusion Policy (DP)** applies diffusion-based generative models to represent and sample robot actions in continuous spaces. 3) **Robotics Decision Transformer (RDT)** (Liu et al., 2025b) adapts the Decision Transformer framework to robotic tasks, enabling goal-conditioned policy learning from offline data. 4) $\pi_0$ (Black et al., 2024) serves as a strong pretrained policy prior that provides generalizable low-level skills across different domains. 5) **GR00T** (Bjorck et al., 2025) is a large-scale, foundation model for robot control trained on diverse multimodal datasets. These models are fine-tuned on the same offline simulator data (FwS indicates Finetuned with Simulated data) generated from our automatic data generation pipeline (Section 4.3) and then deployed zero-shot to real-world applications.

Table 3 presents the evaluation results. Across all six tasks, our method consistently achieves the highest success rates in both simulation and real-world environments, while also completing tasks with fewer average steps. In particular, our method attains an average real-world success rate of 60.8%, significantly outperforming the best baseline with an absolute improvement of over 35%. This substantial margin highlights the robustness and generalization capabilities of our approach, especially under domain shift from simulation to the real world. The performance gap is even more pronounced in complex, long-horizon tasks such as Dual-Arm Water Pouring, Pan Open and Place, and Items Hand-Over and Place, where baseline models frequently fail or struggle to generalize beyond their training distributions. These tasks require temporally extended reasoning and precise coordination, which our method handles reliably, while others often exhibit brittle or erratic behavior during real-world execution. Moreover, another important finding is that while small-scale models like ACT and DP demonstrate competence in short-horizon or low-variance settings, they consistently underperform in more challenging, long-horizon tasks. Their limited model capacity, lack of hierarchical planning, and inability to generalize across diverse tasks hinder their effectiveness in realistic, multi-stage manipulation scenarios. These results collectively demonstrate that our method is not only capable of mastering complex individual tasks but also exhibits the scalability and generalization ability required for real-world, multi-task, long-horizon robotic manipulation, which represents a critical step toward building reliable, general-purpose robotic agents.

Table 3: Robotic manipulation performance (mean ± 95% confidence interval) across different long horizon tasks.

| Tasks | Singe-Arm Water Pouring (200) | | | Dual-Arm Water Pouring (250) | | | Table Rearrangement (250) | | |
|---|---|---|---|---|---|---|---|---|---|
| Methods | Sim. | Real. | Steps | Sim. | Real. | Steps | Sim. | Real. | Steps |
| ACT$_{(FwS)}$ | 6/50±0.09 | 0/20±0.00 | 200.00±0.00 | 6/50±0.09 | 1/20±0.10 | 247.65±4.92 | 2/50±0.05 | 0/20±0.00 | 250.00±0.00 |
| DP$_{(FwS)}$ | 11/50±0.11 | 2/20±0.13 | 199.00±1.63 | 5/50±0.08 | 2/20±0.13 | 247.60±3.72 | 7/50±0.10 | 2/20±0.13 | 246.30±5.33 |
| RDT$_{(FwS)}$ | 33/50±0.13 | 3/20±0.16 | 197.35±3.44 | 21/50±0.14 | 3/20±0.16 | 243.95±6.99 | 18/50±0.13 | 2/20±0.13 | 248.05±3.16 |
| $\pi 0_{(FwS)}$ | 38/50±0.12 | 6/20±0.20 | 194.30±4.20 | 25/50±0.14 | 5/20±0.19 | 241.70±7.72 | 11/50±0.11 | 4/20±0.18 | 237.55±12.41 |
| $\pi 0 - $FAST$_{(FwS)}$ | 31/50±0.13 | 11/20±0.22 | 185.95±8.06 | 30/50±0.14 | 8/20±0.21 | 223.95±15.50 | 23/50±0.14 | 7/20±0.21 | 230.60±12.95 |
| GR00T N1.5$_{(FwS)}$ | 29/50±0.14 | 9/20±0.22 | 189.05±7.05 | 22/50±0.14 | 7/20±0.21 | 231.80±12.81 | 16/50±0.13 | 4/20±0.18 | 237.20±8.87 |
| Sim2Real-VLA | **46/50±0.08** | **17/20±0.16** | **174.60±8.63** | **47/50±0.07** | **16/20±0.18** | **195.15±16.05** | **44/50±0.09** | **16/20±0.18** | **197.05±14.44** |

| Tasks | Items Hand-Over and Place (400) | | | Basket Pick-and-Place (400) | | | Pan Open and Place (550) | | |
|---|---|---|---|---|---|---|---|---|---|
| Methods | Sim. | Real. | Steps | Sim. | Real. | Steps | Sim. | Real. | Steps |
| ACT$_{(FwS)}$ | 0/50±0.00 | 0/20±0.00 | 400.00±0.00 | 0/50±0.00 | 0/20±0.00 | 400.00±0.00 | 0/50±0.00 | 0/20±0.00 | 550.00±0.00 |
| DP$_{(FwS)}$ | 0/50±0.00 | 0/20±0.00 | 400.00±0.00 | 0/50±0.00 | 0/20±0.00 | 400.00±0.00 | 0/50±0.00 | 0/20±0.00 | 550.00±0.00 |
| RDT$_{(FwS)}$ | 8/50±0.10 | 1/20±0.10 | 397.15±5.97 | 12/50±0.12 | 1/20±0.10 | 396.50±7.33 | 15/50±0.13 | 2/20±0.13 | 539.65±15.09 |
| $\pi 0_{(FwS)}$ | 12/50±0.12 | 4/20±0.18 | 388.50±12.17 | 15/50±0.13 | 2/20±0.13 | 395.95±5.83 | 12/50±0.12 | 1/20±0.10 | 546.40±7.53 |
| $\pi 0 - $FAST$_{(FwS)}$ | 10/50±0.11 | 1/20±0.10 | 398.70±2.72 | 13/50±0.12 | 3/20±0.16 | 396.35±4.26 | 11/50±0.11 | 3/20±0.16 | 547.50±5.23 |
| GR00T N1.5$_{(FwS)}$ | 18/50±0.13 | 3/20±0.16 | 395.50±5.33 | 9/50±0.11 | 2/20±0.13 | 397.95±2.95 | 17/50±0.13 | 1/20±0.10 | 545.35±9.73 |
| Sim2Real-VLA | **31/50±0.13** | **8/20±0.21** | **370.20±19.11** | **29/50±0.14** | **9/20±0.22** | **364.15±19.76** | **30/50±0.14** | **7/20±0.21** | **525.35±16.34** |

We conduct a quantitative analysis to evaluate how well Sim2Real-VLA generalizes across different types of configurations in realistic environments. More specifically, we intentionally alter the **original** configuration of the real-world environment to introduce domain gaps, including variations in **background** texture, manipulation **objects'** location, texture, and shape, and the **table**'s surface texture. Besides, we also experiment Sim2Real-VLA robustness to the combination of these gaps. Figure 4 shows the environment configuration at the first run of the experiment. For each domain gap,

we evaluate Sim2Real-VLA; over 20 trials, each with a different sampled configuration, and report the number of successful manipulations.

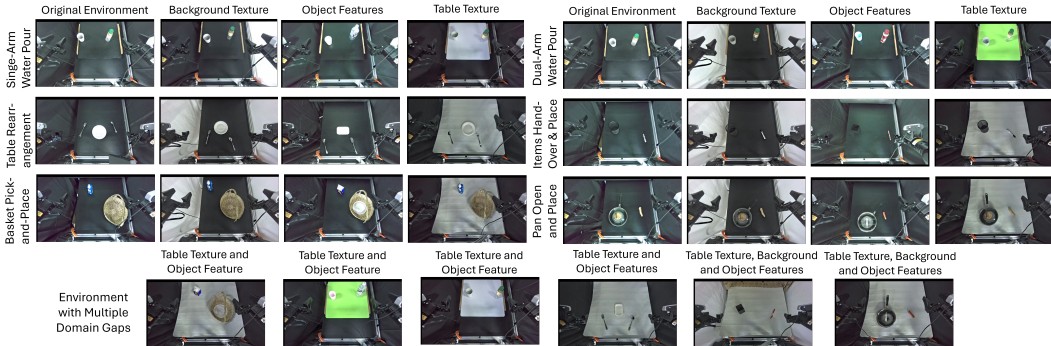

Figure 4: Visualization of environment configurations under the domain gaps of background texture, object features, and table texture across different manipulation tasks.

Table 4 illustrates the generalization ability of Sim2Real-VLA under different domain gaps. Despite variations in background, object properties, and table texture, as well as their combinations Sim2Real-VLA achieves comparable success rates across most tasks. These results indicate that the model maintains stable performance and demonstrates strong robustness to real-world differences. Another intriguing finding is that, for some tasks, performance actually improves under certain domain variations. This effect can be explained by domain shifts reducing spurious correlations present in the original setup, thereby encouraging the policy to rely on task-relevant features. Moreover, some variations may inadvertently simplify specific trials or align more closely with conditions encountered during training, resulting in higher success rates.

Table 4: Number of successful/total trials across different manipulation tasks and domain gaps.

| Task / Domain Gap | Original | Background | Object | Table | Multiple Gaps |
|---|---|---|---|---|---|
| Single-Arm Water Pour | 17/20 | 17/20 | 16/20 | 17/20 | 16/20 (Table + Object) |
| Dual-Arm Water Pour | 16/20 | 16/20 | 16/20 | 17/20 | 17/20 (Table + Object) |
| Table Rearrangement | 16/20 | 15/20 | 14/20 | 16/20 | 15/20 (Table + Object) |
| Item Hand-Over and Place | 8/20 | 9/20 | 8/20 | 6/20 | 8/20 (Table + Object) |
| Basket Pick-and-Place | 9/20 | 9/20 | 10/20 | 9/20 | 7/20 (Table + Background + Object) |
| Pan Open Pick-and-Place | 7/20 | 6/20 | 7/20 | 7/20 | 8/20 (Table + Background + Object) |

**Analyzing Attention Maps in VLA Models** Attention maps serve as a useful diagnostic tool for analyzing how and why performance improves through the incorporation of a chain-of-affordances. By inspecting where the model allocates its attention during action prediction, we gain insights into whether the reasoning is aligned with task-relevant visual and proprioceptive cues.

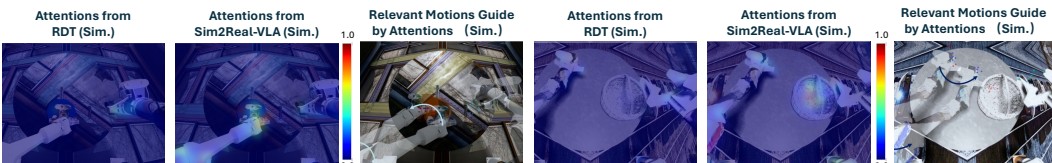

Figure 5: Visualization of attention maps and relevant robot motions during robotic manipulation.

Figure 5 visualizes the attention maps of Sim2Real-VLA's action transformer blocks and compares them against those from a vanilla RDT baseline without affordance integration. The contrast is clear: without the guidance of affordance, the model's attention is broadly distributed, often covering irrelevant background regions, entire objects regardless of their role, or robot joints unrelated to the current manipulation step. In contrast, the affordance-driven Sim2Real-VLA directs its focus to precisely those spatial regions that are critical for the current sub-task. These observations shows that affordances encourage localized attention, ensuring that each action step conditions on the most informative object parts and motion-critical pixels.

## 6 CONLUSION

In this work, we introduced Sim2Real-VLA, an affordance-driven Vision-Language-Action model that achieves zero-shot generalization from exclusively synthetic training data to diverse real-world robotic manipulation tasks. By reformulating manipulation as a structured chain-of-affordances and coupling high-level reasoning with low-level execution through a dual-system architecture, the framework effectively filters out irrelevant variability and focuses on task-critical dynamics. Our automatic data generation pipeline further enables scalable training without manual demonstrations, while extensive experiments across dexterous, bimanual, and long-horizon scenarios demonstrate significant improvements—over 35% higher real-world success rates compared to competitive baselines. Beyond its empirical performance, Sim2Real-VLA highlights the importance of model-side design choices, rather than solely pursuing high-fidelity simulation, for bridging the long-standing Sim2Real gap. *These findings point toward a promising paradigm shift: building robotic foundation models that are trained entirely in simulation, yet are robust to realistic deployment.* Future research will extend our framework to multi-agent collaboration, interactive environments beyond tabletop settings, and the integration of reinforcement learning for continual policy refinement.

## ETHICS STATEMENT

Our study focuses on developing a Vision-Language-Action (VLA) framework for robotic manipulation, trained entirely in simulation and evaluated in controlled real-world environments. We have carefully considered potential ethical concerns in accordance with the requirements of ICLR. Specifically:

- All training data were either automatically generated in simulation environments or sourced from publicly available datasets released for research purposes. No personally identifiable information or human subject data were used at any stage. Real-world demonstrations involved only non-sensitive household objects, ensuring no compromise to privacy or human welfare.
- To minimize potential harm, the proposed framework was developed and rigorously tested under strict monitorty protocols. All real-world evaluations took place in controlled laboratory settings, ensuring that the robot operated within defined safety boundaries and posed no risk to humans, property, or the environment.
- The primary goal of this research is to advance embodied AI systems for beneficial real-world applications, including assistive robotics, resource handling, and safe automation. We explicitly discourage any harmful or malicious use of this technology. Future deployment should adhere to domain-specific safety standards and ethical guidelines to ensure responsible use and positive societal impact.

## REPRODUCIBILITY STATEMENT

We have made extensive efforts to ensure the reproducibility of our results. Details of our Vision-Language-Action (VLA) framework, including model architecture and training procedures, are provided in Section 4.1 and Section 4.2 and further elaborated in Appendix A. We include comprehensive descriptions of all simulation environments, task definitions, and evaluation protocols in Section 5. To facilitate reproducibility, we have submitted anonymized robot manipulation videos as part of the supplementary materials. While we have not included the model parameters and environment source code due to their large size, we will open-source them alongside the code on GitHub upon publication. These resources will enable researchers to replicate our experiments and validate our findings.

## ACKNOWLEDGMENTS

This work is supported in part by Shenzhen Science and Technology Program under grant KJZD20240903104008012, Shenzhen Science and Technology Program under grant ZDCY20250901113000001, CUHK-CUHK(SZ)-GDSTC Joint Collaboration Fund No. 2025A0505000053, GuangDong Key Laboratory of Big Data Computing (2021B1212040002) and Guangdong Provincial Key Laboratory of Mathematical Foundations for Artificial Intelligence (2023B1212010001).

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

## A    DETAILS OF SETTING AND IMPLEMENTATION FOR SIM2REAL-VLA

### A.1    MODEL ARCHITECTURE & KEY PARAMETERS

Regarding the model architecture, we employ DiNOv2 as the visual encoder and T5-XXL as the language encoder. The action expert model comprises approximately 200M parameters, structured with an 8-layer transformer backbone featuring 256 hidden dimensions and 8 attention heads for action processing. The specific archiutecture of our action moodel can be seen in the Figure 6 below. This architecture is complemented by two additional transformer blocks of identical configuration dedicated to affordance inference and guidance, alongside multiple MLP adapters that facilitate dimensional alignment across action, observation, and affordance inputs.

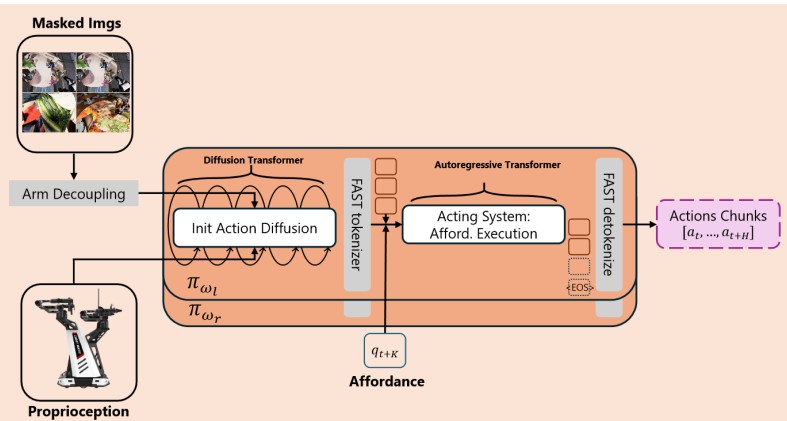

Figure 6: The detailed architecture of action model.

Regarding visual observation masking, the mask prediction module utilizes a standard CNN-based architecture to process raw visual inputs and yield stable object masks. Through the implementation of joint training and domain randomization, the module ensures robust generalization across diverse objects and environmental conditions.

These masked visual observations, combined with language instructions, are subsequently fed into the affordance prediction model. Structured as a regressive transformer, this model outputs a sequence of 2D keypoints projected from key poses into the camera image space as affordances, which effectively serve as high-level plans.

To bridge the affordance-prediction subsystem and the acting policy, the action model is formally constructed as a conditional autoregressive transformer. The pipeline initiates by employing a diffusion-based action expert to generate action trajectories through denoising, conditioned on the aforementioned masked visual observations and proprioceptive inputs. These refined action chunks are tokenized by a pretrained FAST tokenizer and embedded. Utilizing a tokenize-then-concatenate strategy, the model fuses these action embeddings with the predicted affordance outputs.

ly, conditioned on language instructions and visual inputs, the transformer predicts logits for action tokens (or the `<EOS>` token), which are decoded into executable action chunks. Empirical evaluations demonstrate that this binding strategy significantly outperforms alternative architectures.

A pretrained validation model is also needed in affordance chain inferrence. Constructed as a regressive transformer classifier, the validtion modeal takes maksed visual observation and state as input, current target affordance as condation, and output a validation signal to label if the target affordance is achieved.

For the training protocol, we implement a cosine-annealing learning rate schedule with a maximum value of 1e-5 across 40,000 epochs, incorporating exponential moving average (EMA) to enhance training stability. The training configuration utilizes a batch size of 8, requiring approximately 36 GPU hours to complete under these specified conditions.

## A.2    CONFIGURING REWARDS IN VLA MODELS

Note that in robotic manipulation tasks, reward signals are not always explicitly defined; instead, they are often implicitly specified by the task goals themselves. For instance, in widely used robot control frameworks such as Vision-Language-Action (VLA) (Kim et al., 2024), reward information is typically embedded in the language commands $l$ that describe the desired outcome (e.g., "Please close the door"). In such cases, the reward function can be interpreted as $R(s, a) = 1$ if the robot successfully completes the task, and $R(s, a) = 0$ otherwise. When more detailed or nuanced reward structures are needed, AI agents can design sophisticated reward functions (Ma et al., 2024a). These functions are crucial for reinforcement learning (RL), particularly following Supervised Fine-Tuning (SFT) of VLA models.

## A.3    DETAILS ON REAL2SIM DATA PROJECTION

The goal of heterogeneous data projection is to map descriptive observations of real-world tasks into a simulated environment, ensuring that the generated skills remain applicable to the target application. Following the approach in (Liu et al., 2025a), this projection encompasses both scene-level and action-level mappings, as detailed below.

**Scene Projection.** We project the static scene information from the real world into a simulated environment. Inspired by Digital Cousins Dai et al. (2024), we first extract per-object information from input RGB images. Each detected object is then matched to its corresponding "digital cousin"—a visually and functionally similar asset from our simulation dataset. For articulated objects (e.g., drawers, boxes), we further post-process them to create fully interactive simulated counterparts by aligning them with CAD models or synthesized assets. However, in cases where three-view images capture only partial scene information (e.g., occluded object surfaces), or when the retrieved scene fails to semantically align with the real-world context, we leverage a Vision-Language Model (VLM) to identify problematic objects, revise the scene configuration, and regenerate a corrected version using an objective generation model (Zhao et al., 2025). Such a pipeline can be automatically implemented in the simulated environment. The specific prompts used to instruct the VLM for this corrective process are provided in Listing 1 and Listing 2 below.

**Action-Trajectory Projection.** Given either an egocentric video of a human manipulating objects or teleoperated demonstrations performed in the real environment, we project both the actions and object interactions onto robot control signals within a simulated environment. These trajectories capture dynamic motion information and serve as seed demonstrations for downstream automatic skill acquisition.

1) *Robot-Action Projection.* We extract hand movement trajectories from two sources: (1) human hand motion in egocentric videos and (2) teleoperated demonstrations using robotic interfaces. These trajectories are then retargeted to robotic end effectors, such as parallel grippers or dexterous hands, by transforming human or teleoperated hand poses into control signals compatible with the target robot. In this work, we primarily execute manipulations using a gripper as the end effector. Accordingly, two representative fingers are selected as proxies for grasping, and their trajectories are retargeted to drive the open-close motion of the gripper jaws.

2) *Robot-Object Interaction.* To accurately capture the interaction between robot and object, we reconstruct the manipulated objects and their spatial relationship with the operator Liu et al. (2025c).

```
You are a visual-inspection agent responsible for ensuring camera
coverage of specified objects.

Task: {task_description}
Target object list: {object_list}

Instructions:
1. Examine the provided image carefully.
2. For each object in the target list, determine whether the object is
**completely visible** (i.e., no obstruction) in the image.
3. If you find any object that is **obstructed** (partially or fully
blocked from view), output **only** the name(s) of those object(s)
from the list, in a comma-separated list.
4. If no object is obstructed, output "All objects fully visible".

Output format (exact):
Obstructed objects: [object_name1, object_name2, ...]
(or)
All objects fully visible
```

Listing 1: Prompt for visual inspection agent to avoid occlusion by ensuring complete object visibility.

```
You are a visual and task-feasibility evaluator for image-based asset
inspection.

Task: {task_description}
Target object list: {object_list}
Real-world task context image/asset description:
{real_world_image_description}

Instructions:
1. Examine the image carefully to check whether each object from the
target list is captured without obstruction.
   - If any object is obstructed, output the names of those object(s)
from the list.
   - Format: Obstructed objects: [...] or "All objects fully visible".
   - Then proceed to step 2.
2. Assess whether the visual assets (i.e., what is shown in the image)
would realistically support executing the real-world task described
(i.e., {real_world_image_description}).
   - If you determine that the assets are **insufficient** to complete
the real task, identify **which object(s)** do not match realistic
observation (for example: object missing, object appearance
unrealistic, object placement wrong etc.).
   - Output: "Task feasibility: No - issues with [object_name1,
object_name2, ...]"
   - If the assets are sufficient, output: "Task feasibility: Yes".

Combined output format (exact):
Obstructed objects: [object_name1, object_name2, ...]
Task feasibility: Yes
(or)
Obstructed objects: [object_name1, object_name2, ...]
Task feasibility: No - issues with [object_name3, object_name4, ...]
```

Listing 2: Prompt for comprehensive task feasibility evaluation, combining visibility assessment with real-world context alignment.

This involves determining the 3D pose at which the end effector (e.g., hand or gripper) engages with the object (e.g., a cup), ensuring that the projected action trajectory reflects realistic physical interactions. We reconstruct 3D object meshes and poses from sequences of image frames within the demonstration videos or teleoperation logs. These object trajectories are then jointly optimized with the corresponding end-effector trajectories to ensure proper alignment in 3D space. This process is fully automatic and does not require manual intervention. For each frame, the optimization refines the physical contact between objects and end effectors, modeling accurate contact dynamics such as grasp stability and force application. We further smooth and regularize the resulting action trajectories to ensure temporal coherence and realism in the simulated environment.

**Clarification on the Real2Sim prior and "zero-shot" Sim2Real transfer.** In this work, we follow the convention in recent real-to-Sim2Real approaches (Fang et al., 2025; Torne et al., 2024) and use the term zero-shot Sim2Real to indicate that the control policy is learned entirely from simulated experience, without any fine-tuning or gradient-based updates on real-world robot data. Our Real2Sim module consumes a small set of real teleoperation and human video trajectories solely to reconstruct task instances and configure simulation environments (e.g., initial states, camera viewpoints, and target poses). These real trajectories are not used as direct supervision signals for the policy network: the policy is trained only on rendered simulated observations and rewards generated in the reconstructed environments. Thus, Real2Sim acts as a prior over task configurations rather than an additional source of real-robot training data, and our reported results should be interpreted as zero-shot deployment of a policy that has never been updated on real sensor frames or real robot trajectories.

### A.4 DETAILS ON GENERATIVE SCENE SCALING

The goal of generative scene scaling is to bridge the Sim2Real gap and enhance the generalization capabilities of robot policies. Policies trained in simulation often fail when deployed in real-world environments due to domain gaps and distribution shifts between the two domains. We primarily adopt the approach described in (Liu et al., 2025a) for scene-level feature sampling, which provides a good initial range for simulation parameters. However, the kinematic constraints of robot manipulators are typically not fully captured by large language models, leading to sampled scene-level features (e.g., object poses) that may result in kinematically infeasible trajectories.

To achieve more efficient and higher-quality trajectory generation, we implement a robot workspace-aware scene scaling method. First, we develop a robot workspace analyzer that precomputes the reachable workspace region using uniform or Monte Carlo joint position sampling combined with forward kinematics. This analysis provides the complete reachable Cartesian space of the robot end-effector. Subsequently, we perform pose sampling within the intersection of the reachable workspace and the object distribution range obtained from the initial scene-level feature sampling. This approach ensures that generated object poses remain within the robot's kinematic reach, thereby guaranteeing feasible trajectory generation. An example can be seen in the Figure 7.
Beyond object-level constraints, we extend the workspace-aware sampling to robotic-level features. We leverage the same robot workspace analyzer to validate that sampled robot end-effector poses fall within the kinematically reachable region. For camera configuration sampling, including position, orientation, and field of view parameters, we compute the intersection ratio between the camera's view frustum and the robot's reachable workspace volume. By constraining the camera placement such that a substantial portion of its view frustum overlaps with the robot's operational space, we ensure optimal visual coverage of task-relevant regions. This workspace-aware camera positioning strategy enhances the quality of visual observations and improves the robustness of vision-based robotic policies.

### A.5 DETAILS ON AUTOMATIC SKILL ACQUISITION

To construct the dataset used for training Sim2Real-VLA, we designed a multi-stage pipeline for automatic skill acquisition, which combines atomic action primitives with large multi-modal language models (MLLMs). The process unfolds as follows.

**Action Bank Construction.** We first curated a set of atomic actions (e.g., grasp, lift, rotate, etc) that serve as the basic building blocks for manipulation. Each action is implemented with a standardized

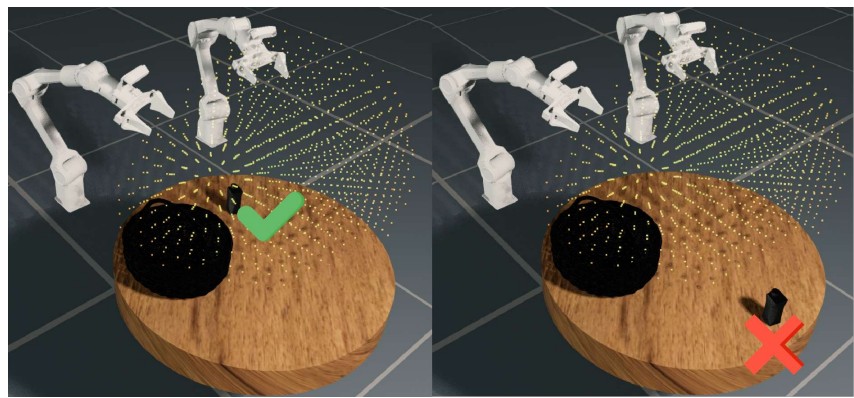

(a) Workspace examination result showcase

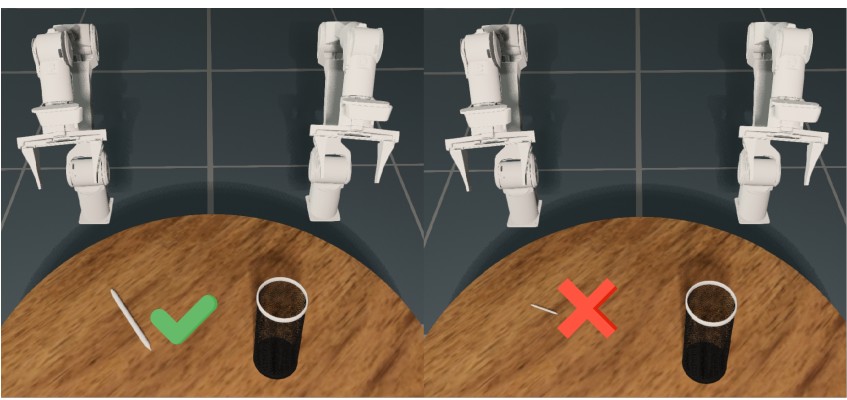

(b) context (scale here, identified by VLM) examination

Figure 7: Succees and fail cases of real2sim projection eamined by workspace analyzer and VLM respectively on asset and action level.

interface, enabling consistent invocation by higher-level planning modules. This collection forms our *Action Bank*.

**Task Decomposition with Task Agent.** For each target task, we employ a multi-modal LLM (GPT-4o in this paper) as a *task agent*. Given a natural-language instruction and visual context, the task agent decomposes the overall objective into a sequence of sub-goals. It generates a step-by-step plan, specifying both the order of execution and the high-level rationale behind each step.

**Action Invocation with Code Agent.** To translate high-level plans into executable robot behaviors, we introduce a second MLLM, the *code agent*. Conditioned on the sub-goals and reasoning generated by the task agent, the code agent selects and invokes the appropriate atomic actions with appropriate configurations (e.g., rotation degrees) from the *Action Bank*. This design separates *task reasoning* from *low-level control*, reducing hallucinations and ensuring that generated action sequences remain grounded in available primitives.

**Key Pose and Trajectory Generation.** Each atomic action is associated with a representative *key pose* of the end-effector (e.g., pre-grasp pose, lifted pose, rotated pose). For each key pose, we apply a generalized inverse kinematics (IK) solver to obtain feasible joint configurations. A trajectory planner then interpolates between consecutive poses, producing smooth and executable motion trajectories.

**Success Condition Generation.** Beyond producing trajectories, it is essential to determine whether a task is completed successfully. We prompt the task agent to propose success conditions for each sub-goal (e.g., *the bottle is lifted above the cup*). These conditions are refined through human feedback and integrated into the pipeline as automatic evaluation signals during data collection.

**Dataset Assembly.** Finally, we store the atomic instructions, joint trajectories, success conditions, detected affordances, and rendered visual observations as structured training samples. This ensures that each data point captures not only robot motion but also the reasoning and affordance information underlying it.

## A.6 SPECIFICATION OF SIMULATION ENGINE

The simulation platform we use is EmbodiChain (Developers, 2025), which is a next-generation robotics simulation and learning platform designed to accelerate research in robot skill acquisition, Sim2Real transfer, and large-scale training. By integrating GPU-accelerated physics simulation, high-fidelity rendering, modular learning environments, and multimodal large language model (MLLM) agents with embodied reasoning capabilities, EmbodiChain provides a unified framework for developing and benchmarking robotic intelligence at scale. Its architecture emphasizes efficiency, realism, and extensibility, enabling researchers to rapidly prototype and evaluate advanced algorithms across diverse tasks and robot morphologies. Below, we detail its core specifications:

**System Architecture.** EmbodiChain is built on a modular, GPU-accelerated framework with three interconnected subsystems:

- *Simulation Engine*: A high-performance rendering and physics backend that supports real-time interaction, large-scale parallelism with extensible APIs, and seamless integration with learning frameworks.

- *Robot Learning Environments*: A suite of standardized, OpenAI Gym-compatible environments with modular functionality for domain randomization, affordable and trajectory generation, reward design, offline dataset collection, online data streaming and more.

- *Embodied Intelligence Framework*: A unified architecture for vision-language-action (VLA) and vision-language model (VLM) design, training, and deployment. It supports both imitation learning from demonstrations and reinforcement learning through environmental interaction, enabling scalable development of multi-modal robotic agents.

## A.7 EMPIRICAL STUDY ON AFFORDANCE CHAIN LENGTH K

Since the proposed method utilizes the affordance chain for reasoning, it is anticipated that the length of the inferred affordance chain significantly influences the overall performance of the model. To quantitatively assess the impact of the inferred affordance chain length, we conduct real-world

experiments across all six tasks, with models trained using affordance chain lengths ranging from $K = 1$ to $K = 3$. The results are presented in Table 5 below.

Table 5: Number of successful/total trials across different manipulation tasks and affordance chain lengths.

| Task / Affordance Chian Length | K=1 | K=2 | K=3 |
|---|---|---|---|
| Single-Arm Water Pour | 17/20 | 10/20 | 11/20 |
| Dual-Arm Water Pour | 16/20 | 11/20 | 8/20 |
| Table Rearrangement | 16/20 | 12/20 | 13/20 |
| Item Hand-Over and Place | 8/20 | 5/20 | 4/20 |
| Basket Pick-and-Place | 9/20 | 9/20 | 4/20 |
| Pan Open Pick-and-Place | 7/20 | 3/20 | 5/20 |

The result above have indicated that, K=1 produced the best performance across all six tasks. We interpret this outcome as indicating that, for our domain, extending the chain beyond one affordance introduces redundancy rather than helpful contextual action guidance.

## A.8 REAL-WORLD EXPERIMENT SETUP

To enhance the reproducibility of our experiment results, we listed several critical initialization details here:

1. Regarding the random seed, we use a seed value of 42 for all random sampling processes.

2. The initial object poses are sampled independently, the distribution for all six tasks is outlined in Table 6, and it mirrors the simulation setup, with the same original reference point located at the base of the robot.

Table 6: Task details including object, xy position, and Z axis rotation.

| Task | Object | XY Position | Z Axis Rotation |
|---|---|---|---|
| Single-Arm Water Pouring | Bottle | [0.67, 0.83], [0.06, 0.22] | [0, 0] |
| | Cup | [0.67, 0.83], [-0.22, -0.06] | [0, 0] |
| Dual-Arm Water Pouring | Bottle | [0.67, 0.83], [0.06, 0.22] | [0, 0] |
| | Cup | [0.67, 0.83], [-0.22, -0.06] | [0, 0] |
| Table Rearrangement | Plate | [0.575, 0.675], [-0.05, 0.05] | [0, 0] |
| | Fork | [0.35, 0.50], [0.11, 0.21] | [-45, +45] |
| | Spoon | [0.35, 0.50], [-0.21, -0.11] | [-45, +45] |
| Items Hand-Over and Place | Pen | [0.52, 0.68], [0.035, 0.195] | [-45, +45] |
| | Holder | [0.5, 0.65], [-0.4, -0.2] | [0, 0] |
| Basket Pick-and-Place | Milk box | [0.81, 0.93], [0.06, 0.22] | [-15, +15] |
| | Basket | [0.65, 0.85], [-0.2, 0.0] | [-15, +15] |
| Pan Open and Place | Pan | [0.4, 0.6], [0.0, 0.2] | [0, 0] |
| | Carrot | [0.51, 0.71], [-0.1, -0.3] | [-15, +15] |

3. The initialization of the robotic arm joint angles for these tasks is also detailed in Table 7, which corresponds to the random initial xpos setup in the simulation, with a range of ±0.02m in xyz direction for all tasks.

Table 7: Initialization of robotic arm joint angles for each task.

| Task | Initial joint (following the parsing order in PhysX) |
|---|---|
| Singe-Arm Water Pouring | [-0.3,0.3,1.0,1.0,-1.2,-1.2,0.0,0.0,0.6,0.6,0.0,0.0,0.0,0.05,0.05,0.05,0.05] |
| Dual-Arm Water Pouring | [-0.3,0.3,1.0,1.0,-1.2,-1.2,0.0,0.0,0.6,0.6,0.0,0.0,0.0,0.05,0.05,0.05,0.05] |
| Table Rearrangement | [-0.15,0.15,1.0,1.0,-1.2,-1.2,0.0,0.0,0.1,2.0,1.2,0.0,0.0,0.0,0.05,0.05,0.05,0.05] |
| Items Hand-Over and Place | [-0.15,0.15,1.0,1.0,-1.2,-1.2,0.0,0.0,0.1,2.0,1.2,0.0,0.0,0.0,0.05,0.05,0.05,0.05] |
| Basket Pick-and-Place | [-0.3,0.3,1.0,1.0,-1.2,-1.2,0.0,0.0,0.6,0.6,0.0,0.0,0.0,0.05,0.05,0.05,0.05] |
| Pan Open and Place | [-0.3,0.3,1.0,1.0,-1.2,-1.2,0.0,0.0,0.1,2.0,1.2,0.0,0.0,0.0,0.05,0.05,0.05,0.05] |

4. The extrinsic parameters of the wrist camera, or more precisely, its relative pose to the attached link, are taken directly from the official URDF of the CobotMagic. For the main binocular camera, calibration is conducted using the CCTag algorithm, yielding an error of 3.8mm.

## A.9 ABLATION STUDY ON ARM-DECOUPLING VS. JOINT LEARNING

To assess the effectiveness of the proposed arm-decoupling design, we conducted an ablation study comparing it with a joint-learning baseline. In the joint-learning setting, a single control module predicted the actions of both arms based on the full visual observation. In contrast, the arm-decoupling design employed two separate control modules, each receiving only the visual feedback associated with its corresponding arm.

We evaluated both models on two representative tasks: (i) a single-arm "Water Pouring" task and (ii) a bimanual "Items Hand-Over and Place" task. For each task, we measured the success rate in simulation, the success rate in the real world, and the average number of control steps required to complete the task. The results are summarized in Table 8.

Table 8: Comparison between the joint-learning baseline and the proposed arm-decoupling design on two manipulation tasks. We report success rates in simulation and the real world, as well as the average number of control steps.

| Method | Single-Arm Water Pouring | | | Items Hand-Over and Place | | |
|---|---|---|---|---|---|---|
| | sim | real | steps | sim | real | steps |
| Joint learning | 0.86 | 0.75 | 178.6 | 0.32 | 0.15 | 390.0 |
| Arm decouple | 0.92 | 0.85 | 174.6 | 0.62 | 0.40 | 370.2 |

As shown in Table 8, the arm-decoupling strategy achieved comparable or slightly better performance than joint learning on the single-arm pouring task in both simulation and real-world settings, while also reducing the average number of control steps. More notably, the arm-decoupling design substantially improved both simulation and real-world success rates for the bimanual hand-over task, together with a reduction in the average number of steps. We interpret these results as evidence that decoupling reduces cross-arm interference: each arm controller can focus on its own relevant visual feedback, thereby avoiding the redundancy and complexity introduced by processing combined wrist-camera observations for simultaneous joint control.

## A.10 FEW-SHOT REAL-WORLD ADAPTATION AND EFFICIENCY

The quantitative results are summarized in Table 9. We analyze the impact of data quantity in Figure 8 and detail the training efficiency in Figure 9.

**Impact of Real Data Quantity (Scaling).** As visualized in Figure 8, baseline methods ($\pi_0$ and $\pi_0^{\text{fast}}$) exhibit a monotonic improvement with increasing real data, relying heavily on demonstrations to correct their poor zero-shot performance. In contrast, our Sim2Real VLA starts with a strong zero-shot baseline (85% on Rearrangement). Notably, we observe a temporary performance dip at 5 demonstrations (dropping to 60%) before recovering to peak performance (90%) at 10 demonstrations. This suggests that a very small amount of real data (5 eps) may initially disrupt the strong simulation prior due to distribution shift ("unlearning" the sim policy), whereas 10 demonstrations are sufficient for the model to effectively adapt and bridge the Sim-to-Real gap.

**Training Dynamics and Efficiency.** Figure 9 (Top Row) details the training curves across different data strategies. While baselines require real data to reach acceptable performance, our method maintains high success rates throughout the training process. We further analyze the cost required to reach these results in the bottom two rows of Figure 9. We estimate FLOPs using the formula from FlashVLA (Tan et al., 2025):

$$\text{FLOPs} = (1 - R) \times \left[ L_p \cdot (4nd^2 + 2n^2d + 2ndm) + (L - L_p) \cdot (4n_pd^2 + 2n_p^2d + 2n_pdm) \right]$$

Our method proves significantly more efficient in both metrics. As shown in Figure 9 (Bottom Row), the Sim2Real VLA converges to high performance in approximately 4 hours, whereas $\pi_0$ requires over 10 hours to achieve comparable results.

Table 9: **Success Rates with Few-Shot Real Data.** Comparison across Sim Only, Real Only (10 demos), and Sim-then-Real (5/10 demos) strategies. Note the non-monotonic behavior ("dip") in our method at 5 eps compared to baselines. Best results per task are **bolded**.

| Model | Data Strategy | Rearrangement | Basket |
|---|---|---|---|
| $\pi_0$ | Sim Only | 0.30 | 0.15 |
| | Real Only (10 eps) | 0.65 | 0.40 |
| | Sim-then-Real (5 eps) | 0.65 | 0.35 |
| | Sim-then-Real (10 eps) | 0.70 | 0.45 |
| $\pi_0^{\text{fast}}$ | Sim Only | 0.40 | 0.25 |
| | Real Only (10 eps) | 0.80 | 0.50 |
| | Sim-then-Real (5 eps) | 0.70 | 0.45 |
| | Sim-then-Real (10 eps) | 0.85 | 0.55 |
| **Ours** | Sim Only | 0.85 | 0.50 |
| | Real Only (10 eps) | 0.75 | 0.40 |
| | Sim-then-Real (5 eps) | 0.60 | 0.35 |
| | Sim-then-Real (10 eps) | **0.90** | **0.60** |

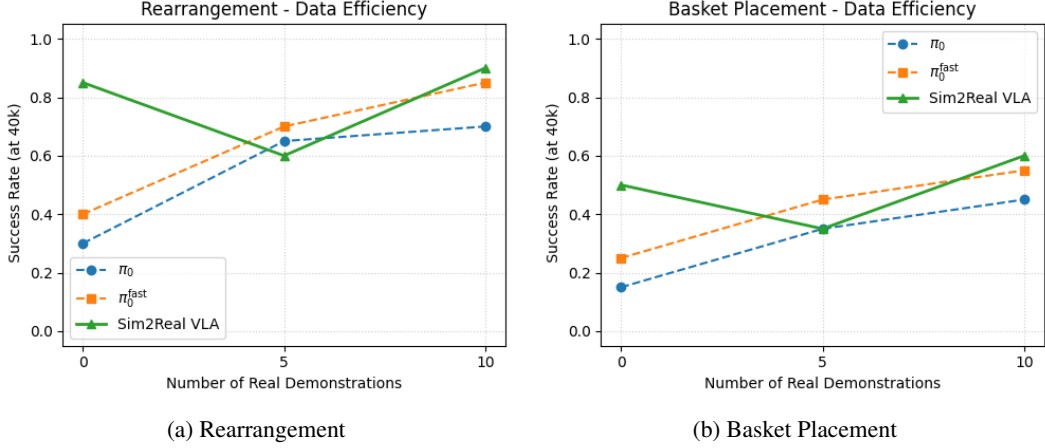

(a) Rearrangement         (b) Basket Placement

Figure 8: **Data Efficiency Scaling.** Success rates (at 40k steps) vs. number of real demonstrations. Baselines improve monotonically. Our method shows a "dip" at 5 eps (due to distribution shift disrupting the sim prior) but recovers to SOTA performance at 10 eps.

## A.11 ROBUSTNESS AND ACCURANCY OF OUR METHOD FACING SIM2REAL PERCEPTION GAP

Our segmentation module is a CNN-based mask-prediction network trained purely on domain-randomized simulation data and jointly optimized with the control policy. The main paper (Table 4) shows that the overall system relying on these masks performs well in real-world manipulation. Here we provide additional quantitative evidence of the robustness and Sim2Real transfer of this segmentation model.

For each of the six manipulation tasks, we first sample 20 observation states from simulator rollouts, together with the corresponding proprioceptive states and predicted masks. We then replay these proprioceptive states on the real robot to collect the corresponding real images and their policy-generated masks. The real images are segmented by SAM, and the SAM masks are downsampled to match the resolution of our model. We compute the mean IoU for two comparisons: (i) real vs. sim masks, and (ii) real vs. SAM masks. The results are summarized in Table 10.
And a example result can be seen in Figure 10 below.

Figure 9: **Analysis of Training Dynamics and Efficiency. (a-b)** Training curves of Sim2Real VLA under different data strategies. The *Sim-then-Real (10 eps)* strategy yields the best final performance. **(c-d)** Success rate vs. compute (TFLOPs). **(e-f)** Success rate vs. wall-clock time (Hours). All efficiency plots (c-f) use the *Sim-then-Real (10 eps)* setting. Our method converges significantly faster (∼4 hours) and with less compute than baselines.

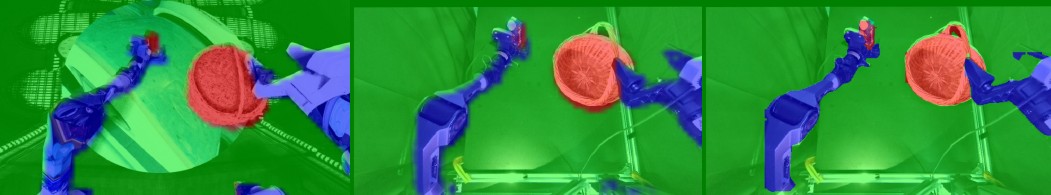

Figure 10: Visualization of predicted mask by segmentation model from our method and SAM, first taken in simulator, the rest two fromthe real-world replay.

To test generalization across sensor configurations and action distributions, we additionally collect 20 teleoperated executions of the single-arm water-pouring task using a camera with different placement and calibration from our standard setup. We apply the segmentation model from the pour-water

Table 10: Mean IoU between segmentation outputs across six tasks. "real vs. sim" compares masks predicted on real vs. simulated images under matched robot states. "real vs. SAM" compares masks predicted on real images with SAM-generated pseudo ground-truth.

| Task | real vs. sim | real vs. SAM |
|------|--------------|--------------|
| Single-Arm Water Pouring | 0.85 | 0.78 |
| Dual-Arm Water Pouring | 0.83 | 0.81 |
| Table Rearrangement | 0.76 | 0.70 |
| Items Hand-Over and Place | 0.78 | 0.75 |
| Basket Pick-and-Place | 0.77 | 0.82 |
| Pan Open and Place | 0.65 | 0.69 |

pretrained checkpoint and compute IoU between its predictions on the new real images and the corresponding SAM masks. The result is reported in Table 11.

Table 11: IoU between segmentation predictions on real images from a different camera setup and SAM masks for the single-arm water-pouring task.

| IoU / Task | Single-Arm Water Pouring |
|------------|--------------------------|
| real vs. SAM | 0.78 |

And a example result can be seen in Figure 11 below.

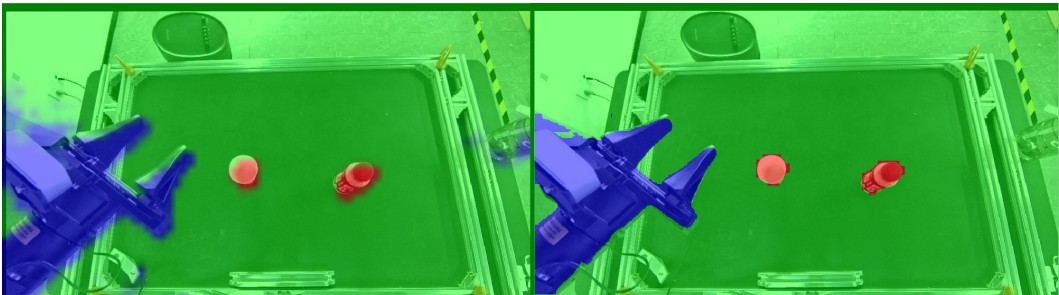

Figure 11: Visualization of predicted mask by segmentation model from our method and SAM on out-of-domain images from a different robot and camera setup.

