# OpenReview forum: "Sim2Real VLA: Zero-Shot Generalization of Synthesized Skills to Realistic Manipulation"
_ICLR.cc/2026/Conference — ICLR 2026 Poster_

### Official Review · Reviewer_84R9 · 2025-10-29

**Soundness:** 2
**Presentation:** 3
**Contribution:** 2
**Rating:** 4
**Confidence:** 4

**Summary:**

The paper proposes Sim2Real-VLA, a dual-system vision-language-action framework trained entirely in simulation that aims to zero-shot transfer to real-world manipulation. A high-level planning system predicts a chain of object-centric affordances from multi-view observations and language, while a low-level acting system executes each affordance with a tokenized action policy; the two stages are connected through object masks and validation signals. The method also introduces (i) object-oriented observation adaptation with domain randomization and (ii) a tokenized action space via DCT→quantization→BPE to compress trajectories. Experiments on six long-horizon/bimanual tasks report an average 60.8% real-world success, >35% absolute over the best baseline using only synthetic training, plus robustness under several domain gaps.

**Strengths:**

The paper offers a clean, end-to-end sim-only training pipeline that links an affordance-conditioned planner with a language-conditioned low-level policy via a tokenized action interface inside a single VLA backbone. While hierarchical affordance→policy decompositions are known, the consolidation of these components into a coherent recipe for zero-shot deployment is well engineered and practically useful. The problem setup and system diagram are clear, and the empirical results show sizable real-robot gains across multiple long-horizon tasks and domain-shift settings. Overall, the contribution is incremental in originality but strong in execution quality and clarity.

**Weaknesses:**

- **Lack comparison to π0-FAST**, Since your “tokenized action space (DCT→quantization→BPE)” is closely aligned with π0-FAST, this comparison is essential to demonstrate benefits beyond tokenization alone.
- **Zero-shot only; lacks a practically meaningful few-shot comparison.**
While zero-shot sim→real is interesting academically, in realistic deployments collecting 5–10 real demonstrations per task is routine and inexpensive. The paper does not evaluate whether the claimed advantages persist when each method is allowed a small, equal real-data budget. Please add a few-shot finetuning study (e.g., 0/5/10 demos per task) comparing your method against π0 and π0-FAST under identical conditions. Report success vs. #demos curves, wall-clock adaptation time, and compute.
- **Unclear advantage over non-VLA affordance→policy pipelines (e.g., AnyGrasp-style)**.
The paper motivates a hierarchical VLA backbone but does not isolate why VLA is needed once an affordance chain is available. A strong control would be a non-VLA baseline: a GraspNet-like affordance estimator feeding a learned policy head (language-conditioned, no VLM/VLA backbone). Compare on the same tasks and sensors, and quantify concrete advantages: (i) robustness to affordance localization errors (error injection tests), (ii) long-horizon credit assignment (success vs. subgoal depth), (iii) compositional generalization to unseen language/object combinations, and (iv) closed-loop latency/stability. Without this, it is difficult to attribute gains to VLA rather than to the affordance decomposition alone.
- Perception at deployment is under-specified. As we know, the perception gap is one of the most challenges in sim-to-real. Object masks are trained in sim with DR, but how robust are they on real images (sensor placement, calibration, segmentation model architecture, failure modes)?
- **“Zero-shot” and Real2Sim prior—scope needs clarification.**
The Real2Sim step maps descriptive observations, including teleoperation and human video trajectories, into sim. Clarify whether any real sensor frames or trajectories directly supervise the policy (vs. only configuring simulation), and ensure the “zero-shot” claim explicitly excludes fine-tuning on real data.

[1] Robust grasping across diverse sensor qualities: The GraspNet-1Billion dataset. IJRR 2023
[2] AnyGrasp: Robust and Efficient Grasp Perception in Spatial and Temporal Domains. TRO 2022

**Questions:**

- Since your “tokenized action space (DCT→quantization→BPE)” is closely aligned with π0-FAST, report the comparison to π0-FAST. Please clarify how your tokenization differs from π0-FAST.
- **Few-shot protocol with π0 / π0-FAST.** Will you include 0/5/10 demo finetuning per task for π0, π0-FAST, and your method with identical training schedules and observation stacks?
- **VLA vs. non-VLA affordance→policy.**
Could you add a baseline that uses your affordance chain but replaces the VLA with a compact policy head (e.g., transformer or MLP) conditioned on language and affordance parameters, trained end-to-end? Please report: (i) zero-shot and 10-demo performance, (ii) sensitivity to synthetic→real mask noise, and (iii) latency/failure-mode breakdowns when subgoal detection drifts.
- **Robustness & recovery from affordance errors.**
How does the controller behave when the affordance predictor is perturbed (e.g., ±3–5 cm pose error, partial/eroded masks)?
- Analysis on perception at deployment. See weakness.
- **“Zero-shot” and Real2Sim prior—scope needs clarification.** See weakness.

---

> ### Author Response · Authors · 2025-11-22
> **Author response to Reviewer 84R9 - Part 1**
>
> Thank you for your thorough review and the critical points you have raised. Your feedback has provided us with a valuable perspective to significantly improve our manuscript. We have carefully addressed all of your concerns in the revised version and provided detailed responses below. We hope the changes we have made will meet with your approval.
>
> > Comment 1 (Weakness 1 & Question 1) Lack comparison to π0-FAST, Since your "tokenized action space (DCT - quantization - BPE)" is closely aligned with π0-FAST, this comparison is essential to demonstrate benefits beyond tokenization alone
>
> >  Since your "tokenized action space (DCT - quantization - BPE)" is closely aligned with π0-FAST, report the comparison to π0-FAST. Please clarify how your tokenization differs from π0-FAST
>
> **Response 1.** Thank you for raising this important point. Since the affordance-guided action module in our proposed method relies on the pretrained FAST tokenizer, it is indeed necessary to compare our approach with Pi0-FAST. We have added the experimental results to Table 2 in the converted version of our paper. As shown in the table, Pi0-FAST trained solely on simulated data performs better on some tasks but worse on others, indicating that its effectiveness varies across different task types. As for details about how the FAST tokenizer is integrated in our model, you can refer to Figure 8 in the revised manuscript and our Response 2 to Reviewer MEpH.
>
> > Comment 2 (Weakness 2 & Question 2) Zero-shot only; lacks a practically meaningful few-shot comparison. While zero-shot sim→real is interesting academically, in realistic deployments collecting 5–10 real demonstrations per task is routine and inexpensive. The paper does not evaluate whether the claimed advantages persist when each method is allowed a small, equal real-data budget. Please add a few-shot finetuning study (e.g., 0/5/10 demos per task) comparing your method against π0 and π0-FAST under identical conditions. Report success vs. #demos curves, wall-clock adaptation time, and compute
>
> **Response 2.** Regarding the few-shot finetuning study, we acknowledge the reviewer’s point that collecting a small amount of real-world data is relatively straightforward. However, as stated in the paper, the central objective of our work is to investigate zero-shot sim-to-real transfer, and our entire architecture is intentionally designed around this motivation.
>
> To address the reviewer’s concern, we conducted additional experiments using 10 real-world demonstrations for both the table-rearrangement and basket pick-and-place tasks. We finetuned our model using these demonstrations while freezing the privileged-prediction module, which relies on simulator-only information. For comparison, we also finetuned pi0 and pi0-FAST under the same real-data setup. Results are available in Appendix A.10 of our updated manuscript.
>
> In summary:
>
>  - Pure real-data finetuning improves the performance of pi0 and pi0-FAST, but their performance remains slightly below or the same as that of our Sim2Real VLA trained purely in simulation.
>
>  - Sim2Real VLA finetuned only on real data underperforms the version finetuned using simulated data. This is expected, as the privileged-module (frozen during real-data finetuning) cannot adapt to the real-world domain shift.
>
> - Simulated Pretrain + Realistic Finetuning (i.e., simulated finetuning followed by real finetuning) consistently outperforms pure real-data finetuning for all baselines and also converges faster, indicating that simulated data is beneficial for leveraging limited real data.
>
>  - Notably, while the sim-to-real finetuned versions of $\pi_0$ and $\pi_0$-FAST significantly outperform their simulation-only counterparts, Sim2Real VLA does not exhibit comparable performance gains. This suggests that additional real-data finetuning is less effective for our specific architecture, likely due to the presence of frozen modules designed to leverage privileged simulator information, which may limit further adaptation.
>
> Overall, these findings reinforce our claim: our method is fundamentally designed for zero-shot sim2real transfer, and while it can benefit from simulated pretraining plus real finetuning, its architecture makes full real-data adaptation less impactful than for baselines.

---

> ### Author Response · Authors · 2025-11-22
> **Author response to Reviewer 84R9 - Part 2**
>
> > Comment 3 (Weakness 3& Question 3) Unclear advantage over non-VLA affordance→policy pipelines (e.g., AnyGrasp-style)
>
> **Response 3.** Thank you for raising this important concern. In our design, the affordance is represented as a set of 2D affordance points obtained by projecting a sequence of poses onto the image space. Their relative offsets are further modulated by the current gripper opening. We believe this representation provides a unified abstraction across heterogeneous sensor configurations (e.g., wrist camera vs. main camera with different resolutions, intrinsics, and extrinsics), allowing the model to better leverage visual input under varying setups.
>
> These 2D affordances are not intended to directly recover a 3D pose. Moreover, our key poses may be object-irrelevant, which is in fact one of the major advantages of our formulation. To highlight this benefit, we intentionally design affordance chains in our tasks that do not correspond to object-centric frames. As a result, AnyGrasp-style object-frame pose prediction pipelines cannot execute these tasks.
>
> For a fair comparison, we train a separate affordance-prediction model on the same dataset, using a loss function that regresses 2D affordance points with an additional term encouraging geometric consistency. We further fine-tune this model on the handover task, which contains a complex sequence of affordances, many of which are unrelated to the object itself. After predicting the affordance points, we follow the industrial practices of keypoint projection, which lifts the keypoints to a 3D pose in the camera frame using a RANSAC PnP-based procedure. After the pose is transformed to the robot frame using camera extrinsics, we then execute the predicted affordance pose.
>
> However, we observe that this pipeline fails to complete the tasks due to accumulated errors in 2D affordance localization, thereby precluding any meaningful assessment of credit assignment, generalization, or latency. We have uploaded a demonstration video at https://anonymous.4open.science/w/Sim2Real-VLA/ , you can refer to it for more details.
>
> These results may suggest that treating object-irrelevant 2D affordance points as intermediate reasoning signals within an end-to-end action-prediction framework is substantially more effective than using them as direct action outputs.
>
> > Comment 4 (Weakness 4 & Question 4) Perception at deployment is under-specified. As we know, the perception gap is one of the most challenges in sim-to-real. Object masks are trained in sim with DR, but how robust are they on real images (sensor placement, calibration, segmentation model architecture, failure modes)?
>
> **Response 4.** We appreciate your insight into this important aspect of sim-to-real transfer. Our segmentation model adopts a standard CNN-based mask-prediction architecture, and—combined with domain randomization and joint training with the policy—can produce stable segmentation results across different objects and background textures, as the results in Table 3. in paper demonstrate that the overall model that relies on its mask output performs well.
>
> To further evaluate the real-world robustness of our segmentation model, we perform the following analysis. For each task, we sample 20 mask predictions from simulator rollouts together with the corresponding proprioceptive states. We then replay these proprioceptive states on the real robot to obtain raw images and their corresponding policy-generated masks. The raw images are segmented by SAM to serve as ground truth, and the SAM masks are downsampled to match the granularity of our segmentation model. We compute the average IoU across six tasks for both real vs. SAM and real vs. sim comparisons. The results are shown in the table below.
>
> | IoU/Tasks | Single-Arm Water Pouring | Dual-Arm Water Pouring | Table Rearrangement | Items Hand-Over and Place | Basket Pick-and-Place | Pan Open and Place |
> | :--- | :--- | :--- | :--- | :--- | :--- | :--- |
> | **real vs sim** | 0.85 | 0.83 | 0.76 | 0.78 | 0.77 | 0.65 |
> | **Real vs SAM** | 0.78 | 0.81 | 0.7 | 0.75 | 0.82 | 0.69 |
>
> To test generalization across sensor setups and actions, we additionally collect 20 teleoperated pouring executions using a camera with different placement and calibration from our standard setup. We apply the segmentation model from the pour-water pretrained checkpoint and compute the IoU between its predictions and SAM. The results can be seen in the table below.
>
> | IoU/Tasks | Single-Arm Water Pouring |
> | :--- | :--- |
> | real vs SAM | 0.78 |
>
> The comparable IoU results indicate that our segmentation model—trained purely on DR-generated data—generalizes well across camera configurations and action distributions. The results above, together with representative in-distribution and out-of-distribution segmentation examples, are available at Appendix 5.10.

---

> ### Author Response · Authors · 2025-11-22
> **Author response to Reviewer 84R9 - Part 3**
>
> > Comment 5 (Weakness 5 & Question 5) "Zero-shot" and Real2Sim prior—scope needs clarification. The Real2Sim step maps descriptive observations, including teleoperation and human video trajectories, into sim. Clarify whether any real sensor frames or trajectories directly supervise the policy (vs. only configuring simulation), and ensure the "zero-shot" claim explicitly excludes fine-tuning on real data
>
> **Response 5.**
>
> We appreciate the reviewer’s thoughtful comment regarding the Real2Sim prior and the scope of our “zero-shot” claim. In the prevailing usage within the robot learning community, it is common to describe a method as achieving zero-shot sim-to-real transfer as long as real-world trajectories are not used to directly train or fine-tune the control policy, even if they are used to synthesize simulated trajectories or configure digital-twin environments, as in ReBot [1] and RialTo [2].
>
> That said, we agree that this terminology can be ambiguous, and your point is well taken. In the revised manuscript, we have added an explicit clarification in Appendix A.3, where we describe our Real2Sim procedure. We now clearly state that (i) real teleoperation and human video trajectories are used only to construct simulation tasks and configure environments, and (ii) the policy is trained exclusively in simulation, without any fine-tuning or other updates on real-world data. We hope this resolves the ambiguity surrounding the “zero-shot” claim. Thank you again for this helpful suggestion.
>
> [1] Fang, Y., Yang, Y., Zhu, X., Zheng, K., Bertasius, G., Szafir, D., & Ding, M. (2025). ReBot: Scaling robot learning with real-to-sim-to-real robotic video synthesis.
>
> [2] Torne, M., Simeonov, A., Li, Z., Chan, A., Chen, T., Gupta, A., & Agrawal, P. (2024). Reconciling reality through simulation: A real-to-sim-to-real approach for robust manipulation.

---

> ### Comment · Reviewer_84R9 · 2025-11-27
>
> Thanks for the authors' thorough rebuttal. Although I still have some concern on why Sim2RealVLA performs worse with 5 demonstrations finetuneing (I think it needs more evidence to support authors' claim about distribution shift), I would like to raise my score considering the quality of whole paper. Thank you.

---

### Official Review · Reviewer_6zd7 · 2025-10-30

**Soundness:** 4
**Presentation:** 3
**Contribution:** 3
**Rating:** 6
**Confidence:** 4

**Summary:**

This paper proposes a Vision-Language-Action model trained entirely in simulation. The method adopts a hierarchical architecture that decouples high-level planning from low-level control execution, and further decomposes tasks into a sequence of affordance-governed atomic subtasks. By leveraging the simulation environment for large-scale data augmentation and affordance extraction, the model mitigates the Sim2Real gap without real-world demonstrations. In the execution module, the authors introduce an arm-decoupled control design for bimanual manipulation, preventing unnecessary coupling and interference between the two arms during task execution. Real-world experiments across multiple long-horizon tasks demonstrate that Sim2Real-VLA significantly outperforms several robotic policy baselines, indicating improved robustness and generalization under domain shift.

**Strengths:**

Overall, the paper presents meaningful contributions in both Sim2Real-oriented algorithmic design and VLA architectural innovation. And the paper is well organized.

**Weaknesses:**

In 4.2, the authors state that a validation model is used to determine whether the current sub-goal has been successfully achieved and whether the system should proceed to the next affordance. Yet, no further explanation is provided regarding its formulation or training procedure. A brief description of this component should be included in the appendix.

In the experimental section, most evaluated tasks are compositions of pick-and-place style atomic tasks. While the results clearly highlight the advantage of hierarchical planning in long-horizon settings, the diversity of atomic-level skills remains limited.

Additionally, further ablation studies would strengthen the work, particularly comparisons isolating the effects of several architecture designs, for example: (i) arm-decoupling versus joint learning, and (ii) the proposed action executing validation mechanism versus predefined goal-based termination conditions.

**Questions:**

Please refer to the Weaknesses part.

---

> ### Author Response · Authors · 2025-11-22
> **Author response to Reviewer  6zd7 - Part 1**
>
> We are grateful for your encouraging assessment and the thoughtful perspective you provided. Your observations have been instrumental in refining our work. We have thoroughly revised the manuscript to reflect your insights and trust that the updated version now meets your expectations.
>
> > Comment 1 (Weakness 1) In 4.2, the authors state that a validation model is used to determine whether the current sub-goal has been successfully achieved and whether the system should proceed to the next affordance. Yet, no further explanation is provided regarding its formulation or training procedure. A brief description of this component should be included in the appendix.
>
> **Response 1.** Thank you for pointing this out. For the validation model, we pretrain a cross-attention transformer that takes as input the current state along with the masked visual-observation affordances. Conditioned on a given affordance, the model outputs whether that affordance has been achieved. During training, observations before an affordance is completed are labeled as not achieved, and the observation at the moment of completion is labeled as achieved.
>
> This pretrained validation model is shared across all six tasks and demonstrates good cross-task generalization. We have added a detailed description of this component to Appendix A.1 of the revised manuscript.
>
> > Comment 2 (Weakness 2)  In the experimental section, most evaluated tasks are compositions of pick-and-place style atomic tasks. While the results clearly highlight the advantage of hierarchical planning in long-horizon settings, the diversity of atomic-level skills remains limited.
>
> **Response 2.** Thank you for raising this concern. While some of the evaluated tasks may appear similar to pick-and-place at a high level, they in fact require a diverse set of atomic skills beyond standard pick-and-place primitives. For example, the pouring-water task involves precise rotational control, where the robot must rotate a specific joint while maintaining stability in the remaining joints. The carrying-basket and opening-pan tasks require part-level manipulation, such as grasping and controlling the basket handle or the pan lid handle. The handover and dual-arm pouring-water tasks further involve coordinated bi-manual manipulation, which introduces additional complexity compared to single-arm operations.
>
> In the meantime, we agree that expanding the diversity of atomic skills is valuable. In future work, we plan to evaluate our system on a broader range of tasks that go beyond pick-and-place–style manipulation. Ongoing progress on tool-use ice scoping tasks with a humanoid robot equipped with a dexterous hand can be found at https://anonymous.4open.science/w/Sim2Real-VLA/.
>
> > Comment 3 (Weakness3) Additionally, further ablation studies would strengthen the work, particularly comparisons isolating the effects of several architecture designs, for example: (i) arm-decoupling versus joint learning, and (ii) the proposed action executing validation mechanism versus predefined goal-based termination conditions.
>
> **Response 3.** Thank you for these insightful suggestions. We agree that further ablation studies clarify the contribution of our architectural choices.
>
> (1). Arm-Decoupling vs. Joint Learning To demonstrate the effectiveness of our arm-decoupling design, we compared our approach against a baseline model trained in a "joint learning" manner (where actions for both arms are predicted by 1 single module with full visual input). We conducted fine-tuning experiments on two representative tasks: the single-arm "pouring water" task and the bimanual "handover" task. The results are presented in the table below.
>
> ### Single-Arm Pouring
> | Method | Sim Success Rate | Real Success Rate | Steps |
> | :--- | :--- | :--- | :--- |
> | **Joint Learning** | 0.86 | 0.75 | 178.6 |
> | **Arm Decouple** | 0.92 | 0.85 | 174.6 |
>
> ### Hand-Over & Place
> | Method | Sim Success Rate | Real Success Rate | Steps |
> | :--- | :--- | :--- | :--- |
> | **Joint Learning** | 0.32 | 0.15 | 390 |
> | **Arm Decouple** | 0.62 | 0.4 | 370.2 |
>
>
> As shown in the table, while the arm-decoupling strategy yields performance comparable to joint learning on single-arm tasks, it significantly improves performance on bimanual tasks. We attribute this improvement to the reduction of interference; by decoupling, each arm's controller focuses only on its relevant visual feedback, avoiding the redundancy and complexity associated with processing combined wrist-camera perception for simultaneous joint control.

---

> ### Author Response · Authors · 2025-11-22
> **Author response to Reviewer 6zd7 - Part 2**
>
> > Comment 3 (Weakness3) Additionally, further ablation studies would strengthen the work, particularly comparisons isolating the effects of several architecture designs, for example: (i) arm-decoupling versus joint learning, and (ii) the proposed action executing validation mechanism versus predefined goal-based termination conditions.
>
> **Response 3.** [ Continued from “Author response to Reviewer 6zd7 - Part 2” ]
>
> (2). Validation Mechanism vs. Predefined Goal-Based Termination:
>  We appreciate the suggestion to compare our approach against a goal-based termination mechanism. However, we respectfully argue that a rigid geometric termination condition is fundamentally incompatible with our 2D affordance-driven framework for two primary reasons:
>
> (a). Incompatibility with 2D Affordance Noise: Our method operates on predicted 2D affordances generated by a neural network, which inherently contain prediction noise. Projecting these 2D points into 3D space to serve as strict geometric termination goals (e.g., coordinate-based thresholds) leads to unreliable performance due to the projection gap. Consequently, a rigid 3D pose rule fails to function effectively within our 2D-driven pipeline, necessitating a learned validation model to robustly infer sub-goal completion.
>
> (b). Dependency on Precise Calibration: Standard goal-based termination relies heavily on privileged information, including specific robot forward kinematics and precise camera extrinsics, to map the end-effector pose between the robot frame and image space. Relying on such strictly calibrated information severely limits the model's ability to generalise to new environments. In contrast, our end-to-end validation model is trained on synthetic data with domain randomisation. This allows it to generalise across diverse sensor configurations and environmental setups without requiring precise recalibration.
>
> For these reasons, we believe that the learned validation mechanism is an essential component of our architecture rather than an optional design choice, making a direct comparison with a rigid, calibration-dependent baseline inequitable.

---

### Official Review · Reviewer_nkJs · 2025-10-30

**Soundness:** 4
**Presentation:** 3
**Contribution:** 3
**Rating:** 6
**Confidence:** 4

**Summary:**

Sim2Real-VLA is a dual-system VLA model trained entirely based on simulation data. It designs a framework in which the high-level planner parses language instructions into a "supply chain", and the low-level executor tracks and verifies segment by segment through tokenized action space. Cooperate with the automated Real2Sim-SceneScaling data generation workflow. This work focuses on innovation on the data side. It first "back-projects" real tasks into simulation, and then uses automatic scaling + skill generation to produce large-scale simulation data. That is, although the method details of Real2Sim are adopted, the overall top-level goal is to bridge the gap between Sim2Real. The experimental scale of the paper is large and the tasks are diverse. It has achieved a success rate improvement of over 35% on real robots for six long-cycle control tasks, providing a new paradigm for zero-shot transfer from pure simulation to the real world. It is suggested that the author explicitly state in one sentence that "Real2Sim is a means and Sim2Real is the end", so as to better eliminate the potential misunderstanding caused by the title "Sim2Real" but the extensive use of Real2Sim in the article.

**Strengths:**

1. Sim2Real-VLA introduces an affordable chain-based design that decomposes complex tasks into verifiable sub-tasks, effectively mitigating error accumulation through a conceptually novel framework.
2. We have developed an automated data generation pipeline comprising real-to-sim projection, scene expansion, and skill generation. This pipeline eliminates the need for manual demonstration and demonstrates significant engineering potential in terms of scalability and large-scale deployment.
3. Extensive real-world experiments were conducted, encompassing single-arm and dual-arm robotic systems, rigid and articulated structures, as well as periodic tasks of varying durations. Furthermore, multiple domain variations—including background environments, object configurations, and tabletop setups—were incorporated, thereby enhancing the robustness and credibility of the evaluation.
4. The study compares against five recent and representative baseline methods, all of which were fine-tuned using identical simulation data to ensure a fair comparison. Additionally, attention visualization is provided to improve the interpretability of the model’s decision-making process.

**Weaknesses:**

1.At the methodological level, the generation mechanism of the affordable chain length K lacks a detailed explanation, with insufficient algorithmic description and absence of statistical distribution analysis. The Real2Sim projection phase does not include a clear failure detection criterion or a defined fallback strategy. Furthermore, the tokenized action space has not been evaluated for quantization errors; the impact of codebook size and sequence length in DCT+BPE on control accuracy remains unexplored, and no reconstruction error analysis or comparisons under high-precision tasks have been provided.
2.At the engineering reproducibility level, critical initialization details are missing in real-world experiments, including random seeds, object initial pose distributions, robotic arm joint angle initialization, and camera extrinsic calibration errors. Additionally, there is no reporting on control frequencies and latency across high-level planning, low-level action token execution, and robot joint controllers. Moreover, fault classification statistics and descriptions of the safety monitoring layer are absent.
3.In the main experimental evaluation, statistical significance analysis is lacking. For the real-world experiment with n=20, it is recommended to report a 95% confidence interval or p-value to assess the reliability of observed performance improvements. Without such metrics, it is unclear whether the reported over 35% improvement stems from systematic gains or random initialization effects. Complementary failure case analyses should also be included to enhance interpretability and robustness assessment.

**Questions:**

1. Is the length K of the affordable chain determined through offline statistical analysis or dynamically predicted in real time? To clarify the generation mechanism and selection strategy for the affordable chain length, could you provide details regarding the decision-making logic, underlying network architecture, and the distribution of K across different tasks?
2. Have there been any instances of failure in the Real2Sim projection within the current framework? If so, what is the observed failure rate? In cases involving unsolvable inverse kinematics (IK) or semantic mismatches, how is the rollback mechanism triggered? It would be helpful to include representative failure cases along with comparative analyses before and after corrective actions.
3. For future work, would it be possible to release the initialization configuration files used in real-world experiments—such as random seeds, object pose sampling ranges, initial joint angles, and camera extrinsic parameters? Additionally, could you report the control frequencies and measured latency values for high-level affordance reasoning, low-level action token generation, and each component of the robot joint controller? Such information would significantly enhance the reproducibility and engineering applicability of the proposed method.
4. To strengthen the statistical validity of the results presented in Table 2 of the main experimental section, please include a 95% confidence interval or an equivalent measure of statistical significance. This would support the claim of a 35% performance improvement and ensure robust interpretation of the findings.

---

> ### Author Response · Authors · 2025-11-22
> **Author response to Reviewer nkJs - Part 1**
>
> Thank you for your positive assessment and for recognizing the value in our work. We are also grateful for the constructive suggestions you provided. We have carefully addressed each of your points in our revisions and hope that our responses have thoroughly resolved your concerns.
>
> > Comment 1 (Weakness 1 & Question 1) At the methodological level, the generation mechanism of the affordable chain length K lacks a detailed explanation, with insufficient algorithmic description and absence of statistical distribution analysis.
>
> > Is the length K of the affordable chain determined through offline statistical analysis or dynamically predicted in real time? To clarify the generation mechanism and selection strategy for the affordable chain length, could you provide details regarding the decision-making logic, underlying network architecture, and the distribution of K across different tasks?
>
> **Response 1.**     Thank you for pointing out the need for greater clarity on the details of the affordance chain length K. Below, we provide a detailed response addressing each of your questions:
>
> Regarding Decision‑making logic and statistical selection of K， in our experiments, we evaluated fixed values of K in the range K=1 to K=3. Empirically, K=1 produced the best performance across all six tasks. We interpret this outcome as indicating that, for our domain, extending the chain beyond one affordance introduces redundancy rather than helpful contextual action guidance. Hence, the chain length was determined offline via statistical analysis rather than being dynamically predicted at runtime. The table below shows the result, which is also updated in Appendix A.7 of the paper.
>
> | | Singe-Arm Water Pouring | Dual-Arm Water Pouring | Table Rearrangement | Items Hand-Over and Place | Basket Pick-and-Place | Pan Open and Place |
> | :--- | :--- | :--- | :--- | :--- | :--- | :--- |
> | **K=1** | **17/20** | **16/20** | **16/20** | **8/20** | **9/20** | **7/20** |
> | **K=2** | 10/20 | 11/20 | 12/20 | 5/20 | 9/20 | 3/20 |
> | **K=3** | 11/20 | 8/20 | 13/20 | 4/20 | 4/20 | 5/20 |
>
> As for underlying network architecture，our affordance–prediction module is implemented as a transformer model. It takes as input masked image tokens and language prompt features. The output consists of a set of 2D affordance points (projected into image space) which then guide action generation. This architecture is described in Appendix A.1, and we have moved a summarized schematic into Figure 7 of the main text per your suggestion.
>
> > Comment 2. (Weakness 2 & Question 2)  The Real2Sim projection phase does not include a clear failure detection criterion or a defined fallback strategy.
> > Have there been any instances of failure in the Real2Sim projection within the current framework? If so, what is the observed failure rate? In cases involving unsolvable inverse kinematics (IK) or semantic mismatches, how is the rollback mechanism triggered? It would be helpful to include representative failure cases along with comparative analyses before and after corrective actions.
>
> **Response 2.** Thank you for your insightful questions regarding failure detection in the Real2Sim projection process. Below, we provide a detailed explanation of our failure detection mechanism and how we handle failure cases:
>
> (1). Failure detection criterion and fallback strategy:
>
> As detailed in Appendix A.5, we utilize a Vision-Language Model (VLM) to detect object occlusions and task‑feasibility issues during the Real2Sim projection phase. The VLM is prompted to identify potential problems with the scene configuration (e.g., occluded object surfaces or semantically mismatched assets). When obstructions are detected, we sample a new initial object location. In cases where an asset is found to be unsuitable or mismatched, we regenerate the asset via a new asset generation call. We have updated the prompt we are using in the Appendix A.5.
>
> On the action level, we employ a workspace analyzer to detect issues in compatible pose sampling. If an action is predicted to result in inverse kinematics (IK) failure (due to an invalid pose), the system triggers a rollback by sampling a new initial pose.
>
> (2). Failure rates:
>
> Based on our experiments across all tasks, we observe that the failure rate related to asset is approximately less than 1%, which is hardly seen in our current settings. While the failure rate due to action-related issues (such as incompatible poses or IK failures) is around 30%.
>
> (3). Failure mode example:
>
> Regarding the failure mode example, we have updated 2 failure case examples (1 for asset level, 1 for action level) in Figure 8. The reviewer can refer to it for more details.

---

> ### Author Response · Authors · 2025-11-22
> **Author response to Reviewer nkJs - Part 2**
>
> > Comment 3. (Weakness 3) Furthermore, the tokenized action space has not been evaluated for quantization errors; the impact of codebook size and sequence length in DCT+BPE on control accuracy remains unexplored, and no reconstruction error analysis or comparisons under high-precision tasks have been provided.
>
> **Response 3.**
>
> (1). As for FAST utilization, we use a pretrained FAST tokenizer in the affordance execution module in an encoder-decoder paradigm. After the initial action chunk is generated from the action diffusion model, it's tokenized by the pretrained FAST tokenizer, then it's embedded and concatenated with the affordance, then it's processed by a transformer model in an autoregressive manner. An extra <EOS> token is added during training to mark the end of the output FAST logits, after which the generated action tokens are extracted, decoded, and the L2 loss is calculated between the output and the ground truth action chunk.
>
> (2) Regarding reconstruction error measuring, we collected 100 (32,32)  action chunks from each fine-tune dataset, and calculated their reconstruction error by the MSE metric of decoded action tokens and the input action chunks, and the result is listed in the table below. As we can see in the table, the reconstruction error caused by quantization is relatively low, at the 1e-5 level. The higher reconstruction error in the Items Hand-Over and Place, Basket Pick-and-Place, Pan Open and Place tasks may be due to the complicated frequency patterns in these long-horizon tasks.
>
> | tasks | Single-Arm Water Pouring | Dual-Arm Water Pouring | Table Rearrangement | Items Hand-Over and Place | Basket Pick-and-Place | Pan Open and Place |
> | :--- | :--- | :--- | :--- | :--- | :--- | :--- |
> | **Reconstruction error** | 6.26 x 1e-5 | 8.37 x 1e-5 | 5.29 x 1e-5 | 1.67 x 1e-4 | 1.24 x 1e-4 | 2.07 x 1e-4 |
> | **Sequence length** | 53.06 | 49.27 | 47.75 | 62.27 | 53.64 | 58.14 |
>
> (3). Regarding the impact of code book size, we utilize a pretrained FAST tokenizer and thus have to keep the same code book size of 1024. (or in other words, changing the code book size manually is useless as the training has completed.)
>
> (4). When it comes to the impact of sequence length, though changing it can make a difference in encoding and decoding, it ought to be the same to align with the pretrained vocabularies. We alter the round scale exponentially, following Figure 12 of the FAST paper, and calculate the reconstruction error of 100 action chunks sampled from all the finetune datasets, only to find that the reconstruction error soars correspondingly. The result can also be seen in the table below.
>
> | Round scale | 0.1 | 1 | 10 (default) | 100 | 1000 |
> | :--- | :--- | :--- | :--- | :--- | :--- |
> | **Reconstruction error** | 0.27 | 4.2 x 1e-3 | 8.9 x 1e-5 | 0.053 | 0.34 |
> | **Sequence length** | 9.34 | 21.73 | 40.13 | 120.64 | 277.34 |

---

> ### Author Response · Authors · 2025-11-22
> **Author response to Reviewer nkJs - Part 3**
>
> > Comment 4 (Weakness 4 & Question 3) At the engineering reproducibility level, critical initialization details are missing in real-world experiments, including random seeds, object initial pose distributions, robotic arm joint angle initialization, and camera extrinsic calibration errors. Additionally, there is no reporting on control frequencies and latency across high-level planning, low-level action token execution, and robot joint controllers.
>
> > At the engineering reproducibility level, critical initialization details are missing in real-world experiments, including random seeds, object initial pose distributions, robotic arm joint angle initialization, and camera extrinsic calibration errors. Additionally, there is no reporting on control frequencies and latency across high-level planning, low-level action token execution, and robot joint controllers.
>
> **Response 4.**     Thank you for raising this important point.
>
> (1). Regarding the random seed, we use a seed value of 42 for all random sampling processes.
>
> (2). The initial object poses are sampled independently, the distribution for all six tasks is outlined in the real-world experiment table below， and it mirrors the simulation setup, with the same original reference point located at the base of the robot.
>
> | task | Object | xy position | Z axis rotation |
> | :--- | :--- | :--- | :--- |
> | **Single-Arm Water Pouring** | bottle | [0.67, 0.83] [0.06, 0.22] | |
> | | cup | [0.67, 0.83] [-0.22, -0.06] | |
> | **Dual-Arm Water Pouring** | Bottle | [0.67, 0.83] [0.06, 0.22] | |
> | | cup | [0.67, 0.83] [-0.22, -0.06] | |
> | **Table Rearrangement** | plate | [0.575, 0.675] [-0.05, 0.05] | |
> | | fork | [0.35, 0.50] [0.11, 0.21] | [-45, 45] |
> | | spoon | [0.35, 0.50] [-0.21, -0.11] | [-45, 45] |
> | **Items Hand-Over and Place** | pen | [0.52, 0.68] [0.035, 0.195] | [-45, 45] |
> | | holder | [0.5, 0.65] [-0.4, -0.2] | |
> | **Basket Pick-and-Place** | Milk box | [0.81, 0.93] [0.06, 0.22] | [-45, 45] |
> | | basket | [0.65, 0.85] [-0.2, 0.0] | [-15, 15] |
> | **Pan Open and Place** | pan | [0.4, 0.6] [0.0, 0.2] | |
> | | carrot | [0.51, 0.71] [-0.1, -0.3] | [-45, 45] |
>
> (3). The initialization of the robotic arm joint angles for these tasks is also detailed in the table below, which corresponds to the random initial xpos setup in the simulation, with a range of ±0.02m in xyz direction for all tasks.
>
> | task | Initial joint (following the parsing order in PhysX) |
> | :--- | :--- |
> | Single-Arm Water Pouring | [-0.3, 0.3, 1.0, 1.0, -1.2, -1.2, 0.0, 0.0, 0.6, 0.6, 0.0, 0.0, 0.05, 0.05, 0.05, 0.05] |
> | Dual-Arm Water Pouring | [-0.3, 0.3, 1.0, 1.0, -1.2, -1.2, 0.0, 0.0, 0.6, 0.6, 0.0, 0.0, 0.05, 0.05, 0.05, 0.05] |
> | Table Rearrangement | [-0.15, 0.15, 1.0, 1.0, -1.2, -1.2, 0.0, 0.0, 1.2, 1.2, 0.0, 0.0, 0.05, 0.05, 0.05, 0.05] |
> | Items Hand-Over and Place | [-0.15, 0.15, 1.0, 1.0, -1.2, -1.2, 0.0, 0.0, 1.2, 1.2, 0.0, 0.0, 0.05, 0.05, 0.05, 0.05] |
> | Basket Pick-and-Place | [-0.3, 0.3, 1.0, 1.0, -1.2, -1.2, 0.0, 0.0, 0.6, 0.6, 0.0, 0.0, 0.05, 0.05, 0.05, 0.05] |
> | Pan Open and Place | [-0.3, 0.3, 1.0, 1.0, -1.2, -1.2, 0.0, 0.0, 1.2, 1.2, 0.0, 0.0, 0.05, 0.05, 0.05, 0.05] |
>
> (4).  The extrinsic parameters of the wrist camera, or more precisely, its relative pose to the attached link, are taken directly from the official URDF of the CobotMagic. For the main binocular camera, calibration is conducted using the CCTag algorithm, yielding an error of 3.8mm.
>
> (5),  The control frequency is set at the default 40 Hz of CobotMagic across all tasks, while the latency between high-level planning and low-level action token execution is approximately 200ms, running on an NVIDIA GeForce RTX 4060 GPU. Latency between low-level action token execution and robot joint controllers is minimal, operating within several milliseconds as it is passed through ROS topics. These details have been updated in Appendix A.8 of the revised  paper.
>
> > Comment 5 (Weakness 5)  Moreover, fault classification statistics and descriptions of the safety monitoring layer are absent.
>
> **Response 5.** Thank you for pointing this out. We do not incorporate additional safety monitoring layers in our setup. Instead, safety during the experiments is ensured through the use of an Emergency Stop (E-Stop) button. If the robotic arm moves into an unsafe position, a human operator can immediately press the E-Stop button to halt the system. This testing is then recognized as a failure in our statistics reported in the tables.

---

> ### Author Response · Authors · 2025-11-22
> **Author response to Reviewer nkJs - Part 4**
>
> > Comment 6 (Weakness 6 & Question 4) For the real-world experiment with n=20, it is recommended to report a 95% confidence interval or p-value to assess the reliability of observed performance improvements.
>
> > To strengthen the statistical validity of the results presented in Table 2 of the main experimental section, please include a 95% confidence interval or an equivalent measure of statistical significance.
>
> **Response 6.**  Thank you for raising this concern. We have conducted statistical analysis across 20 trials to calculate a 95% confidence interval for each task and updated the results in Table 1 of the revised version of the paper.

---

> > ### Comment · Reviewer_nkJs · 2025-11-26
> >
> > The authors have addressed all my concerns, and I’m happy to raise my score.

---

### Official Review · Reviewer_MEpH · 2025-10-31

**Soundness:** 3
**Presentation:** 2
**Contribution:** 3
**Rating:** 8
**Confidence:** 2

**Summary:**

In this work, the authors propose Sim2Real-VLA, a Vision-Language-Action (VLA) model that is trained on synthetic data but can be applied to real-world manipulation tasks. The method consists of two components: a high-level Affordance Prediction model and a low-level Action system. The proposed approach outperforms existing baselines.

**Strengths:**

The authors tackle the relevant problem of sim2real transfer. They propose a novel approach that uses a Vision-Language-Action (VLA) model to generate affordances and structure the chain of affordances. This is an innovative idea that, in combination with the low-level Acting system, makes the proposed method effective while maintaining architectural simplicity.

**Weaknesses:**

In the main part of the paper, several important aspects are missing or insufficiently explained and should be clarified to improve clarity and reproducibility.

First, the simulation engine is only mentioned in Appendix A6 as EmbodiChain. However, there is no reference provided, and no public information about this engine appears to be available. Given the importance of the simulator in the training pipeline, the authors should include a proper citation or at least a brief technical description in the main text.

Second, while some details of the VLA model are provided in the appendix, the paper would benefit substantially from a more detailed presentation of these components in the main body. This would help readers better understand how the model operates and how it integrates with the affordance prediction and action systems.

Overall, the proposed method is interesting and demonstrates promising results. I am inclined to recommend acceptance at this stage. However, I would like to emphasize that without sufficient information about the simulator and the data generation process, I would be inclined to lower the score due to concerns regarding reproducibility and transparency.


Minor Remarks
	•	Line 68: There is an extra blank space (“Sim2Real-VLA . By coupling”).
	•	Line 206: Domain randomization is reintroduced, though it was already mentioned in line 201.
	•	Line 272: The authors mention Byte-Pair Encoding (BPE) but do not provide a citation or further explanation. A reference or an equivalent description would improve clarity.

**Questions:**

Please see Weaknesses.

Have the authors considered evaluating their approach on different robot embodiments to assess generalization across hardware platforms?

Have the authors considered testing the method on deformable object manipulation tasks to further demonstrate robustness and versatility?

---

> ### Author Response · Authors · 2025-11-22
> **Author response to Reviewer MEpH - Part 1**
>
> We are deeply grateful for your positive assessment and for recognizing the merits of our work. Thank you for your encouraging words and thoughtful suggestions. We have taken all of your feedback to heart and provide detailed responses below. We sincerely hope that these clarifications have thoroughly addressed your concerns.
> > Comment 1. (Weakness 1) First, the simulation engine is only mentioned in Appendix A6 as EmbodiChain. However, there is no reference provided, and no public information about this engine appears to be available. Given the importance of the simulator in the training pipeline, the authors should include a proper citation or at least a brief technical description in the main text.
>
> **Response 1.** Thank you for pointing out that the simulation engine plays a critical role in our sim‑to‑real policy training pipeline. In our work, we follow the data‑generation strategy of DexScale (cited in Section x of the manuscript) for constructing the simulated data pipeline. With respect to the simulator itself, we use EmbodiChain as mentioned in Appendix A.6, with open source code in https://github.com/DexForce/EmbodiChain, a framework built on in-house renderer and PhysX. This setup allows us to achieve both photorealistic visual observations and precise physics‐based interaction during skill synthesis. Hope these further citations can provide you with sufficient information about the simulator and address your concern.
>
> > Comment 2. (Weakness 2) Second, while some details of the VLA model are provided in the appendix, the paper would benefit substantially from a more detailed presentation of these components in the main body. This would help readers better understand how the model operates and how it integrates with the affordance prediction and action systems.
>
> **Response 2.**  To address the comment regarding the integration of the affordance‑prediction subsystem and the acting system, we clarify that our pipeline proceeds as follows: First, raw action trajectories are denoised using a diffusion‑based action expert, conditioned on masked visual observations and proprioceptive input. Subsequently, the resulting action chunks are tokenized using the pretrained FAST tokenizer and embedded. We concatenate these embeddings with the predicted affordance outputs to create a combined representation. This representation is fed into an autoregressive transformer, which predicts logits for action tokens or the \<EOS\> token, conditioned on masked visual observations and language instructions. Lastly, the predicted tokens are decoded back into action chunks. To illustrate this process, we have updated the architecture diagram in Figure 7 of the revised manuscript. Empirically, we found that this “tokenize‑then‑concatenate” affordance-action binding strategy outperforms alternative model architectures
>
> > Comment 3. (Weakness 3)  Minor Remarks:
>  > 1. Line 68: There is an extra blank space ("Sim2Real-VLA . By coupling").
>  > 2. Line 206: Domain randomization is reintroduced, though it was already mentioned in line 201.
>
> **Response 3.** We appreciate you pointing out these issues. We have removed the extra blank space on Line 68. Regarding the mention of domain randomization on Line 206, we clarify that this passage serves as a detailed elaboration on our factor selection strategy and specifies the exact parameters modified, building upon the general introduction provided in Line 201.
>
> > Comment 4. (Weakness 4) Line 272: The authors mention Byte-Pair Encoding (BPE) but do not provide a citation or further explanation. A reference or an equivalent description would improve clarity.
>
> **Response 4.**     Thank you for highlighting this oversight.
> BPE[1] is a sub‑word tokenisation algorithm originally from data compression and widely used in NLP to merge the most frequent adjacent symbol pairs into new tokens until a desired vocabulary size is reached.
>
> In our pipeline, the pretrained FAST[2] tokenizer applies this idea to robotics action chunks: it first applies a discrete cosine transform (DCT) and quantizes the resulting vectors to integers, then a BPE‐style code book encodes the most frequently occurring integer pairs into higher‑level tokens. Through million‑scale pre‑training of action chunks, this representation supports efficient learning in our affordance‑guided action model.
>
> We have inseredt a citation of the original BPE algorithm at Line 272 in the revised manuscript, and FAST has been cited at Line 272.
>
> [1] Gage, P. (1994). A new algorithm for data compression. The C Users Journal, 12(2), 23-38.
>
> [2] Pertsch, K., Stachowicz, K., Ichter, B., Driess, D., Nair, S., Vuong, Q., ... & Levine, S. (2025). Fast: Efficient action tokenization for vision-language-action models. arXiv preprint arXiv:2501.09747.

---

> ### Author Response · Authors · 2025-11-22
> **Author response to Reviewer MEpH - Part 2**
>
> > Comment 5. (Question 1 & 2)
> > Question 1: Have the authors considered evaluating their approach on different robot embodiments to assess generalization across hardware platforms?
> > Question 2: Have the authors considered testing the method on deformable object manipulation tasks to further demonstrate robustness and versatility?
>
> **Response 5.**    We appreciate the reviewer raising the interrelated questions regarding cross‑embodiment generalization and deformable‑object manipulation. To address the hardware adaptability, we selected a dual‑arm cobot Magic platform specifically because this setup allows us to encompass a comprehensive spectrum of embodiment configurations. By designing six tasks that include single‑arm pouring, dual‑arm pouring, and handover operations, our evaluation effectively covers distinct manipulation modes ranging from independent single‑arm actuation to coordinated dual‑arm interaction. Furthermore, to demonstrate generalization across both distinct morphologies and object properties, we are currently extending our evaluation to a humanoid robot equipped with dexterous hands. These experiments target deformable manipulation tasks, such as dexterous doll grasping. We also note that our simulator (EmbodiChain, built on PhysX) natively supports soft bodies and volume deformables, providing a robust foundation for these tasks. While formal quantitative analysis is ongoing, we have provided preliminary videos on https://anonymous.4open.science/w/Sim2Real-VLA/.

---

### Author Response · Authors · 2025-12-02
**Summary of updates**

We sincerely thank the AC and the reviewers for their time and thoughtful feedback. Below we (i) explain **how the response and revision address the main concerns** and (ii) briefly summarize the **strengths** highlighted in the reviews.

We have substantially addressed the concerns along **three directions**:

1. **Model Architecture and Key Component Details of Proposed Method**: We provided comprehensive details on the overall model architecture, explicitly outlining the realization of the mask prediction, affordance prediction, action model, and validation model. Specifically, we added a diagram Figure 7 to elucidate the action model's architecture, demonstrating the integration of predicted affordances into action generation. Furthermore, we clarified the 2D keypoint representation, the decision logic for the affordance chain length $K$, and the integration of FAST tokenizers within our framework. All the updated details above are demonstrated in A.1.

2. **Further Experiments including Baseline Comparison, Ablations, Parameters Impact, Few-Shot Manner**: We strengthened the empirical evaluation by incorporating the $\pi_0$-FAST baseline to isolate the benefits of our architecture beyond FAST tokenization. We also underscored the crucialness of the VLA backbone through comparisons with a non-VLA modular baseline prone to geometric error accumulation. We conducted ablation studies on the Arm Decouple feature across single and dual-arm tasks to verify its necessity, alongside sensitivity analyses for the affordance chain length K and FAST tokenizer parameters. Additionally, we verified that our performance advantage persists when all methods are restricted to an equal, few-shot real-data budget. Finally, we demonstrated the robustness of the simulation-trained mask predictor against OOD inputs to bridge the Sim2Real gap. All the additional experiments results can be seen in the Appendix (A.7 for affordance chain length $K$, A.9 for ablation study on arm decoupling, A.10 for few-shot study, A.11 for sim2real mask gap.).

3. **Infrastructure, Terminology, Experiments, and References Clarification**: We explicitly detailed the "Real2Sim" projection used for data generation in Appendix A.5 and clarified that our usage of "zero-shot" refers to the absence of real-world training demonstrations in Appendix A.3, consistent with prior literature. We also enhanced reproducibility by including full experimental configurations in Appendix A.8 and updated the manuscript with necessary citations for the FAST tokenizer and BPE.

Recognized strengths of the work (as noted by the reviewers)

1. **Novel and effective dual-system architecture**: The proposed hierarchical framework, combining a high-level affordance model as planner with a low-level action model as actor, effectively filters irrelevant features and handles long-horizon tasks. (Reviewer MEpH, 6zd7, 84R9)

2. **Strong zero-shot Sim2Real generalization**: The method demonstrates impressive capability in transferring synthesized skills to real-world manipulation tasks without manual fine-tuning. (Reviewers MEpH, nkJs, 84R9)

3. **Scalable and automated data generation**: The integration with an automated real2sim-enhanced data synthesis pipeline is recognized as a significant advantage for scaling up manipulation skills. (Reviewer MEpH, nkJs)

4. **Promising results on complex tasks**: The experiments successfully validate the method on diverse and challenging scenarios, including bimanual and long-horizon manipulation. (Reviewer 6zd7, 84R9)

According to the feedback posted by the reviewers, **Reviewer nkJs indicated that our rebuttal successfully addressed their concerns and subsequently raised the score**. Similarly, **Reviewer 84R9 confirmed an increase in their score based on the overall quality of our work**. Although the other two reviewers, including Reviewer MEpH (initial score 8) and Reviewer 6zd7 (initial score 6), have not replied due to an issue with the OpenReview system, we believe that our responses have also effectively addressed their concerns, and we remain confident that they are positively inclined toward accepting our article.

We hope this summary helps the AC in evaluating the revised manuscript.

---

### Meta-Review · Area_Chair_7QTY · 2026-01-07

**Summary:**

This submission receives an initial score of 8664. After rebuttal, the authors resolve concerns regarding details of methods and experiments and comparisons with more baselines and ablations.  The concerns of Reviewer 84R9 (score 4) are mostly resolved. Thus, my decision is acceptance.

**Reviewer Concerns:**

Concerns of details of methods and experiments and comparisons with more baselines and ablations are all solved.

**Reviewer Scores:**

Reviewer 84R9 might raise the score to positive as most of the concerns are solved.

Other reviewers' might stay positive scores.

---

### Decision · Program_Chairs · 2026-01-26

Accept (Poster)